# Recombinational branch migration by the RadA/Sms paralog of RecA in *Escherichia coli*

**Deani L Cooper[1,2], Susan T Lovett[1,2]***

[1]Department of Biology, Brandeis University, Waltham, United States; [2]Rosenstiel Basic Medical Sciences Research Center, Brandeis University, Waltham, United States

**Abstract** RadA (also known as 'Sms') is a highly conserved protein, found in almost all eubacteria and plants, with sequence similarity to the RecA strand exchange protein and a role in homologous recombination. We investigate here the biochemical properties of the *E. coli* RadA protein and several mutant forms. RadA is a DNA-dependent ATPase, a DNA-binding protein and can stimulate the branch migration phase of RecA-mediated strand transfer reactions. RadA cannot mediate synaptic pairing between homologous DNA molecules but can drive branch migration to extend the region of heteroduplex DNA, even without RecA. Unlike other branch migration factors RecG and RuvAB, RadA stimulates branch migration within the context of the RecA filament, in the direction of RecA-mediated strand exchange. We propose that RadA-mediated branch migration aids recombination by allowing the 3' invading strand to be incorporated into heteroduplex DNA and to be extended by DNA polymerases.

***For correspondence:** lovett@brandeis.edu

**Competing interests:** The authors declare that no competing interests exist.

## Introduction

All organisms have complex mechanisms to accurately replicate and repair their chromosomes to maintain genetic integrity. In *E. coli*, the RecA protein promotes repair of DNA lesions directly through its role in homologous recombination (reviewed in [*Persky and Lovett, 2008*]). In addition, it promotes repair indirectly by the recruitment of repair polymerases to damaged replication forks (*Patel et al., 2010*) and by activation of the SOS response, a transcriptional response to DNA damage (reviewed in [*Simmons et al., 2009*]). Each of these processes depends on the formation of RecA filaments on single-strand DNA (ssDNA).

In vitro RecA mediates strand exchange, a key step of recombination, in three distinct phases (*Radding et al., 1983*). The first phase is the formation of the presynaptic filament on ssDNA. RecA filaments form when dimers nucleate on DNA in a slow step (*Bell et al., 2012*); subsequently the filament is extended in both directions, although at a higher rate at the 3' end of the RecA:ssDNA filament. When ssDNA is bound in the primary DNA binding site of the RecA filament, it is underwound relative to B-form dsDNA such that the RecA-DNA filament has about 18 bases per turn (*Chen et al., 2008*; *Galletto et al., 2006*). ATP binding, but not hydrolysis, is required for active RecA filament formation. The second phase involves the homology search process and strand-pairing in which duplex DNA is bound and sampled for pairing through a secondary DNA binding site (*Mazin and Kowalczykowski, 1996*, *1998*). After homologous DNA molecules are paired, the third phase of strand exchange involves branch migration, in which the region of heteroduplex DNA formed between the two strand exchange partners is extended. The heteroduplex is initially bound through RecA primary site interactions, with the displaced strand(s) in the secondary site (*Mazin and*

**eLife digest** Damage to the DNA of a cell can cause serious harm, and so cells have several ways in which they can repair DNA. Most of these processes rely on the fact that each of the two strands that make up a DNA molecule can be used as a template to build the other strand. However, this is not possible if both strands of the DNA break in the same place. This form of damage can be repaired in a process called homologous recombination, which uses an identical copy of the broken DNA molecule to repair the broken strands. As a result, this process can only occur during cell division shortly after a cell has duplicated its DNA.

One important step of homologous recombination is called strand exchange. This involves one of the broken strands swapping places with part of the equivalent strand in the intact DNA molecule. To do so, the strands of the intact DNA molecule separate in the region that will be used for the repair, and the broken strand can then use the other non-broken DNA strand as a template to replace any missing sections of DNA. The region of the intact DNA molecule where the strands need to separate often grows during this process: this is known as branch migration. In bacteria, a protein called RecA plays a fundamental role in controlling strand exchange, but there are other, similar proteins whose roles in homologous recombination are less well known.

Cooper and Lovett have now purified one of these proteins, called RadA, from the *Escherichia coli* species of bacteriato study how it affects homologous recombination. This revealed that RadA can bind to single-stranded DNA and stimulate branch migration to increase the rate of homologous recombination. Further investigation revealed that RadA allows branch migration to occur even when RecA is missing, but that RadA is unable to begin strand exchange if RecA is not present. The process of branch migration stabilizes the DNA molecules during homologous recombination and may also allow the repaired DNA strand to engage the machinery that copies DNA.

Cooper and Lovett also used genetic techniques to alter the structure of specific regions of RadA and found out which parts of the protein affect the ability of RadA to stimulate branch migration. Future challenges are to find out what effect RadA has on the structure of RecA and how RadA promotes branch migration.

*Kowalczykowski, 1998*). ATP hydrolysis is required for complete heteroduplex product formation when the homologous molecules exceed several kilobases in length (*Jain et al., 1994*).

Several proteins modulate the RecA filament, regulating the recombination activity of RecA and its potentially mutagenic activity resulting from induction of the SOS response. These accessory proteins include those involved in RecA nucleation onto ssDNA (PsiB), in loading and unloading of RecA onto DNA coated with single-strand DNA binding protein (SSB) (RecFOR) and in regulating RecA filament stability (DinI and RecX) (reviewed in *Cox (2007)*). Other proteins such as UvrD (*Petrova et al., 2015*), PcrA (*Fagerburg et al., 2012*; *Park et al., 2010*), and RuvAB (*Eggleston et al., 1997*), and DinD dismantle RecA filaments (*Uranga et al., 2011*).

In eukaryotes, there are several paralogs of the major recombination protein Rad51 that either modulate Rad51 activity or are specialized strand-exchange proteins themselves (*Adelman and Boulton, 2010*; *Bernstein et al., 2013*; *Gasior et al., 2001*; *Qing et al., 2011*; *Suwaki et al., 2011*; *Taylor et al., 2015*). In bacteria, there is at least one partially characterized RecA paralog, RadA. RadA (also known as 'Sms', for 'sensitivity to methyl methanesulfonate' [*Song and Sargentini, 1996*] was identified as a radiation-sensitive mutant of *E. coli* [*Diver et al., 1982a*]) and is required for DNA recombination and repair in many diverse bacterial species (*Beam et al., 2002*; *Burghout et al., 2007*; *Carrasco et al., 2004*; *Castellanos and Romero, 2009*; *Cooper et al., 2015*; *Krüger et al., 1997*; *Lovett, 2006*; *Slade et al., 2009*). Thus, RadA is a possible candidate for a RecA accessory protein. Despite its name, RadA of eubacteria is not orthologous to RadA of archaea, the latter being a true strand-exchange protein, functionally and structurally similar to bacterial RecA and eukaryotic Rad51 (*Seitz et al., 1998*; *Wu et al., 2004*; *Yang et al., 2001*).

In *E. coli,* RadA affects recombination measured by certain in vivo assays, often in a manner partially redundant to other functions that mediate late steps of homologous recombination. Loss of *radA,* by itself, reduces recovery of genetic rearrangements at tandem-repeated sequences, which

are promoted by defects in the replication fork helicase, DnaB (*Lovett, 2006*). In addition, loss of RadA reduces homologous recombination when in combination with loss of RuvAB or RecG (*Beam et al., 2002*), as measured by conjugation with Hfr donors. RuvAB and RecG are DNA motor proteins that branch-migrate recombination intermediates such as Holliday junctions during the late stages of homologous recombination (reviewed in (*Persky and Lovett, 2008*). For sensitivity to genotoxins including azidothymidine (AZT), ciprofloxacin (CPX) and UV irradiation, mutations in *radA* are especially strongly synergistic with those in *recG* (*Beam et al., 2002*; *Cooper et al., 2015*). This genotoxin-sensitivity in *radA recG* strains appears to be due, in part, to the accumulation of recombination intermediates, since it can be suppressed by early blocks to recombination (by mutations in *recF* or *recA*) or by overexpression of RuvAB (*Cooper et al., 2015*). These genetic observations implicate RadA in late steps of recombination, potentially involving branch migration of recombination intermediate DNA structures such as Holliday junctions.

RadA is a 460 amino acid protein that has three well-conserved domains found in other proteins, as well as a 5-amino acid motif highly conserved among *radA* orthologs. The N-terminal 30 amino acids form a putative zinc-finger domain with a C4 motif, CXXC-X$_n$-CXXC. In bacteria, proteins with this domain include the DNA repair proteins UvrA and RecR, and the ATP-dependent serine protease ClpX. *The E. coli radA100* mutation, a C28Y mutation in the putative Zn finger motif, negates *radA* function in vivo and is partially dominant (*Cooper et al., 2015*; *Diver et al., 1982b*). The second RadA domain (aa 59–184) is homologous to the ATPase region of RecA and contains both Walker A and Walker B boxes and regions homologous to its L1 and L2 loops involved in primary site DNA binding. A RadA-K108R mutation at the Walker A sequence is a dominant-negative RadA allele in *E. coli*  (*Cooper et al., 2015*). The C-terminal 150 amino acids comprise a predicted S5 domain 2-type fold, (EMBL-EBI Interpro subgroup IPR014721, http://www.ebi.ac.uk/interpro/entry/IPR014721), present in ribosomal proteins S5 and S9, EF-G, Lon, RNase P, MutL, and several DNA topoisomerases. Deletion of this domain negates RadA functions in vivo (*Cooper et al., 2015*). In BLAST alignments, this region is most closely related to the ATP-dependent serine protease Lon (*Chung and Goldberg, 1981*). Mutation of serine 372 of RadA, comparable in alignments to the active site serine of Lon, did not affect RadA genetic function and this serine is not conserved among RadAs; this and the lack of other conserved residues of the Lon protease catalytic triad indicate that RadA is unlikely to possess serine protease activity. Between the RecA and S5 domain 2 domains, there is a conserved motif specific to RadA proteins, KNRFG, a motif also found in the phage 29 structure-specific nuclease (*Giri et al., 2009*). The K258A mutation in this motif negates RadA function and is partially dominant in vivo (*Cooper et al., 2015*).

To explore RadA function in *E. coli*, we purified wild type RadA as well as several site-directed mutants altered in conserved motifs of the protein. We found that the wild-type RadA protein preferentially binds single-strand DNA in the presence of ADP, exhibits ATPase activity stimulated by DNA, and increases the rate of RecA-mediated recombination in vitro by stimulation of branch migration. Branch migration can be mediated by RadA even in the absence of RecA and is highly directional in nature, with preferential extension of the heteroduplex in the 5' to 3' direction, relative to the initiating single-strand; this is codirectional with that of RecA-mediated strand exchange. Mutations in the Walker A, KNRFG and zinc finger motifs abolish RadA's branch migration activity in RecA-coupled reactions and lead to the accumulation of strand exchange intermediate species. The ability of RadA to catalyze branch migration in the context of the RecA filament and its codirectionality with strand exchange distinguish it from other branch migration functions in *E. coli*, RecG and RuvAB (*Whitby et al., 1993*). RadA's ability to branch migrate recombination intermediates readily explains *radA* mutant phenotypes in vivo.

## Results

### Protein purification

To elucidate RadA structure and function, we purified native wild-type RadA and several RadA domain mutants and then evaluated their biochemical activities, particularly those possessed by the RecA protein. Wild-type RadA was estimated to be more than 98% pure (*Figure 1—figure supplement 1*).

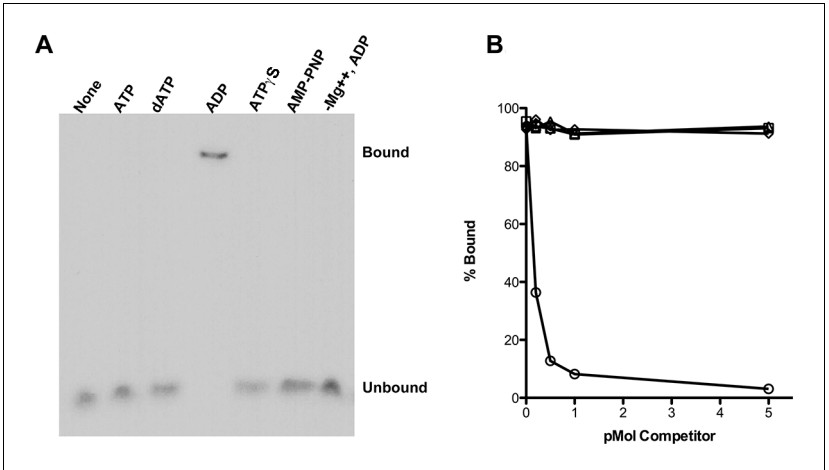

**Figure 1.** RadA Binding to DNA. (**A**) Nucleotide dependence of RadA binding to poly d(T)$_{30.}$ Reactions (10 µl) contained 100 fmol (molecule) of radio-labeled poly d(T)$_{30,}$ 3.3 pmol RadA and 1 mM nucleotide. After incubation at 37 °C for 20 min, binding was assessed using EMSA. (**B**) DNA substrate specificity of RadA Binding. Reactions containing 1 mM ADP, 3.3 pmol RadA, 100 fmol (molecule) of radio-labeled poly d(T)$_{30}$ and unlabeled competitor DNA (circles-poly d(T)$_{30,}$ triangles- poly d(C)$_{30,}$ diamonds-poly d(G)$_{30,}$ squares poly d(A)$_{33}$ were incubated at 37° for 20 min. The extent of binding was determined using scanned autoradiographs of EMSA gels processed with Image J-64.

The following figure supplements are available for figure 1:

**Figure supplement 1.** Purification of RadA.

**Figure supplement 2.** RadA binding to poly(dT)$_{30}$.

**Figure supplement 3.** RadA binding curve to poly(dT)$_{30}$.

**Figure supplement 4.** Binding of RadA to substrate E2.

## DNA binding

Using an electrophoretic mobility shift assay (*Figure 1A*), we examined the binding of purified RadA to poly(dT)$_{30}$ in the presence of various nucleotide cofactors. RadA bound this oligonucleotide only in the presence of ADP. No binding was detected in the absence of nucleotide, with ATP or dATP or in the presence of poorly-hydrolyzable ATP analogs, ATPγS or AMP-PNP (*Figure 1A*). These observations suggest that ADP promotes the most stably DNA-bound RadA. This behavior contrasts to that of RecA, which requires ATP for binding to form the active, extended DNA filament and which dissociates in the presence of ADP. Poorly hydrolyzable nucleotide analogs such as ATPγS produce the most stable RecA binding (*McEntee et al., 1981*).

Competition experiments (*Figure 1B*) with unlabeled poly(dA), poly(dC), poly(dG) and poly(dT) showed that only poly(dT) competes for RadA binding to labeled poly(dT), indicating a binding preference for by poly(dT). RecA protein also shows a preference for poly(dT) (*Bugreeva et al., 2005*; *McEntee et al., 1981*), which may be because poly(dT) assumes a more flexible structure (*Mills et al., 1999*). The apparent K$_D$ of RadA binding to poly(dT)$_{30}$ is about 110 nM, with a Hill coefficient of 1.5 (*Figure 1—figure supplement 2*, *Figure 1—figure supplement 3*), indicative of cooperative binding. RadA was observed to bind poly(dT)$_{30}$ when flanked on both 5' and 3' ends by 30 nucleotides of natural DNA sequence (substrate 'E2', *Figure 1—figure supplement 4*); this binding was inhibited if poly(dA)$_{30}$ was allowed to anneal to the substrate, showing that RadA binds more poorly to duplex DNA.

## DNA-stimulated ATPase

Using an NADH-coupled assay (*Table 1*), we measured the ATPase activity of RadA, in the presence or absence of various DNA cofactors, circular ssDNA (φX174 virion) and dsDNA (φX174 RF DNA). Like RecA, RadA's ATPase activity is strongly stimulated by ssDNA. However, RadA's ATPase is also substantially stimulated by dsDNA whereas dsDNA stimulates RecA's ATPase only after a lag period (*Kowalczykowski et al., 1987*). RadA's ATPase activity in the presence of ssDNA was measured to have a kcat of 29.4 min$^{-1}$, comparable to that of RecA (*Weinstock et al., 1981*). Addition of single-strand DNA-binding (SSB) protein after incubation of RadA or RecA with ssDNA was observed to repress the ATPase activity of RadA, whereas it slightly stimulated the ATPase of RecA (*Figure 2*). This latter observation suggests that, although RecA exhibits stable binding to ssDNA when challenged by SSB, less stable binding by RadA allows for SSB competition and inhibition of its ATPase activity. RadA ATPase activity measured with a variety of different DNA structures formed by oligonucleotides, including forks, splays, and other branched structures, showed no significant difference from that with ssDNA (*Figure 2—figure supplement 1*). Although these DNA molecules are only very weakly bound by RadA in gel shift experiments (data not shown), they are sufficient to stimulate RadA's ATPase activity, indicating some transient or unstable association.

We purified several forms of RadA, mutated in its characteristic motifs: C28Y (Zn finger), K108R (Walker A box), K258A (KNRFG RadA motif) and S372A (putative Lon protease active site) (*Figure 3A*, *Figure 3—figure supplement 1*). These were analyzed for poly(dT)$_{30}$ binding in the presence of ADP and ssDNA-dependent ATPase activity (using the more sensitive PEI TLC method) (*Figure 3B,C*). As expected, the K108R mutant in the Walker A box abolished ATPase activity; a defect in ATPase was exhibited by the K258A mutant as well. Both of these mutants were defective for DNA binding in the presence of ADP. RadA mutant C28Y retained the ATPase activity but was defective in DNA binding. RadA mutant S372A retained ATPase and DNA-binding activities.

## Strand exchange reactions

We examined standard RecA-mediated strand-exchange reactions between 5386 nucleotide circular φX174 ssDNA and linear duplex DNA in the presence of ATP and an ATP-regeneration system (*Figure 4A*). Presynaptic filaments are formed by the incubation of RecA with ssDNA, linear dsDNA is then added and the reaction is initiated by the addition of ATP and SSB. Reactions are monitored by agarose gel electrophoresis. In this regimen, RecA catalyzes the formation of branched DNA molecules within 5 min, which are converted to the relaxed circular dsDNA final product by 18 min. Addition of RadA, at a 1:17 stoichiometry relative to RecA, accelerated final product formation, such that the reaction was complete by 5 min, with no detectable accumulation of branched intermediates (*Figure 4B*). Although RadA did not change the efficiency of the RecA strand-exchange reaction, RadA enhanced the branch migration phase of the reaction to yield the final product more quickly. This branch migration was directional in nature and drove the reaction forward to the nicked circular product rather than back to the linear substrate.

The strand exchange reaction was also performed by initiating the reaction with RecA alone and adding RadA to the ongoing reaction after 5 min, when all the duplex linear DNA had been converted to branched intermediates (*Figure 4B*). In these reactions, final product is visible as early as

**Table 1.** DNA dependence of RadA ATP hydrolysis.

| DNA substrate | Apparent kcat (ATP/RadA/min) |
| --- | --- |
| None | 2.9 +/- 0.2 |
| Circular single-strand | 29.4 +/- 0.4 |
| Supercoiled double-strand | 9.1 +/- 0.4 |
| Linear double-strand | 4.2 +/- 0.2 |
| Nicked double-strand | 15.2 +/- 0.5 |

Reactions contained 1 mM ATP and 20.3 µM (nucleotide) φX174 DNA. The values shown are the average of two experiments, except for the circular single-strand value. It is the average of five experiments. Standard deviations are reported.

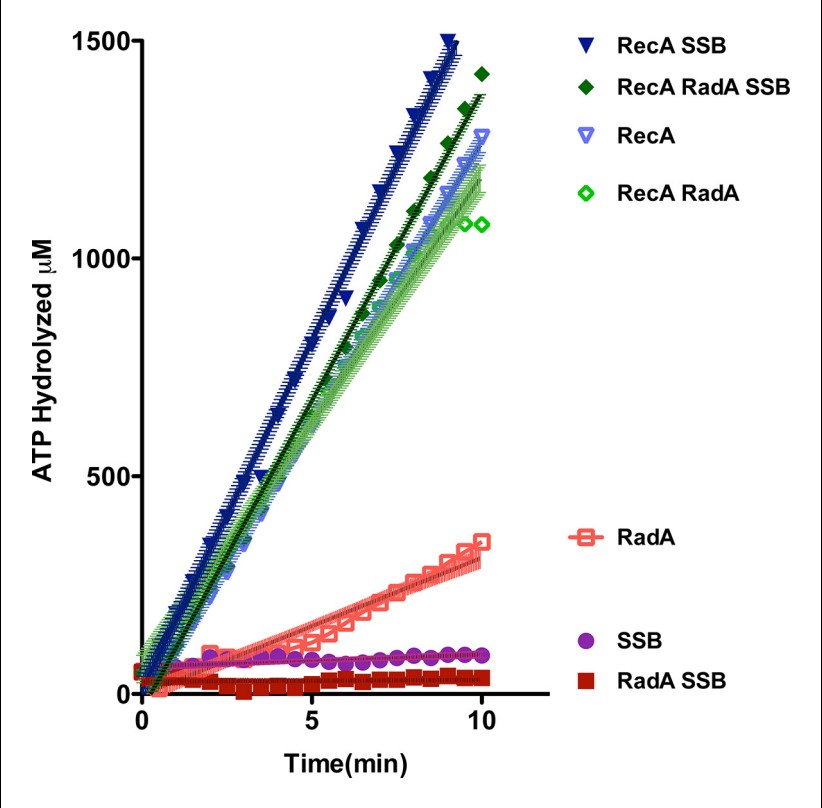

**Figure 2.** ATP hydrolysis in reactions including RecA, SSB and RadA. ATP hydrolysis was measured in reactions containing DNA and protein concentrations similar to those in recombination reactions and included 21 µM (nucleotide) single-strand circular DNA and 6.7 µM RecA, 1.9 µM SSB, and 630 nM RadA. RecA and/or RadA were pre-incubated with the single-strand DNA for 8 min at 37 °C before the reactions were initiated with ATP +/- SSB. Rate measurement started 5 min after the addition of ATP. Reactions included: SSB (closed purple circles), RadA (open red squares), RadA +SSB (closed dark red squares), RecA alone (blue inverted triangles), RecA+SSB (closed dark blue inverted triangles), RadA + RecA (open green diamonds), RadA+RecA+SSB (closed dark green diamonds). Error bars represent the 95% confidence interval of the linear fit of the data calculated using Prism Graph Pad.

The following figure supplement is available for figure 2:

**Figure supplement 1.** ATPase activity on model oligonucleotide substrates.

2.5 min after RadA addition, with the reaction complete after an additional 5 min. Therefore, RadA need not be present in the reaction during RecA presynaptic filament formation and can stimulate branch migration from RecA-promoted strand-exchange intermediates. Order-of-addition experiments (*Figure 4—figure supplement 1*) measuring RecA-mediated strand exchange, with and without RadA, showed that RadA strongly stimulated branch migration when added before SSB, but not when added after. In addition, RadA did not detectably stimulate RecA-strand exchange in the absence of SSB. RadA stimulated RecA-mediated strand exchange reactions when RecA concentrations were reduced to suboptimal levels but did not when RecA became limiting (*Figure 4—figure supplement 2*). When RecA was held at saturating concentration (6.7 µM) and RadA was titrated, the stimulation of branch migration was detectable at RadA:RecA stoichiometries of as little as 1:223 and reached the maximal detectable stimulation at 1:13 (*Figure 4—figure supplement 3*). In the RadA-stimulated reactions, the time of appearance of final product varied with RadA concentration.

In the absence of an ATP-regeneration system, the addition of ADP to the 3-strand reaction inhibits strand exchange, primarily by destabilizing the RecA presynaptic filament (*Kahn and Radding, 1984*; *Lee and Cox, 1990a*; *1990b*; *Piechura et al., 2015*; *Wu et al., 1982*). As an indication of how RadA might affect the RecA filament, we assayed ADP-inhibition of the standard RecA 3-strand

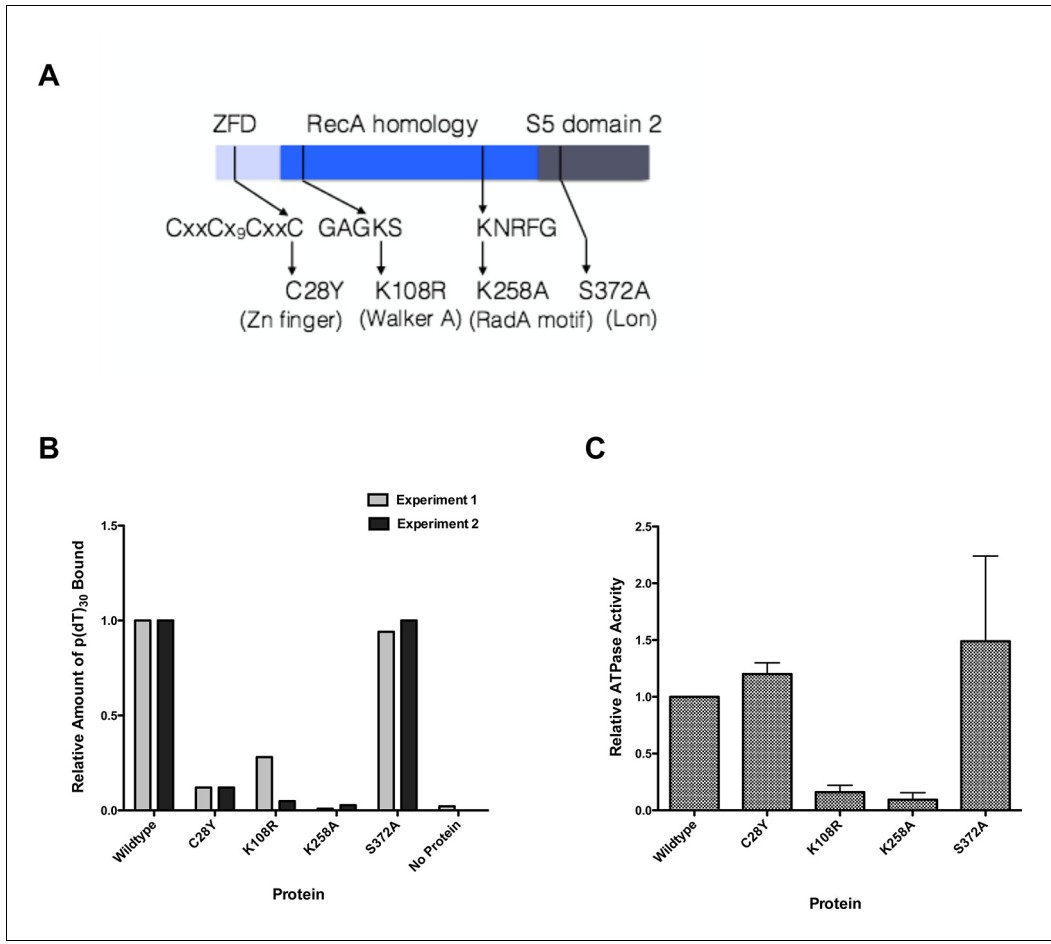

**Figure 3.** Properties of RadA Domain Mutants. (**A**) Schematic of RadA Domains and Locations of Domain Mutants. (**B**) Binding of RadA Mutants to poly d(T)$_{30}$. Reactions contained 100 fmol (molecule) of poly d(T)$_{30,}$ 3.3 pmol (mutant) RadA and 1 mM ADP. After incubation at 37 °C for 20 min, binding was assessed using EMSA. Binding relative to wild-type RadA is shown for two independent experiments. (**C**) ATP hydrolysis by RadA domain mutants. ATP hydrolysis was assessed using PEI TLC to visualize release of inorganic phosphate (Pi) as described in the procedures. Reactions contained either no or 10.5 µM (nucleotide) single-strand circular M13 DNA and 250 nM RadA or RadA mutant protein. Graph shows the mean ATP hydrolysis activity of RadA mutants from three independent experiments relative to the activity of wild-type RadA. Error bars represent the standard deviation of the mean.

The following figure supplement is available for figure 3:

**Figure supplement 1.** SDS-PAGE of RadA mutants.

reaction (lacking an ATP-regenerating system), at various concentrations relative to a fixed amount of ATP, 5 mM (*Figure 4C*). In 60 min reactions with RecA alone, ADP completely inhibited strand-exchange at 4 mM. In reactions containing both RecA and RadA, ADP inhibition of strand exchange was almost complete at 1 mM and complete at 2 mM. Furthermore, even without ADP addition, RadA inhibited RecA 3-strand transfer reactions that lack an ATP-regeneration system. We note that under these reaction conditions, which include SSB, RadA's ATPase is inhibited (*Figure 2*), so this result is not simply due to a higher rate of ATP depletion by the mere addition of RadA. Under the conditions used for RecA presynaptic filament formation, addition of RadA at 1:17 RadA:RecA caused a slight reduction in the measured ATPase (*Figure 2*), which appears to be primarily due to RecA. This may indicate some removal and replacement of RecA with RadA on ssDNA, or alternatively, that RadA, at substoichiometric amounts, can inhibit RecA's ATPase while bound to ssDNA. RadA did not promote RecA strand transfer or joint molecule formation when SSB was omitted from

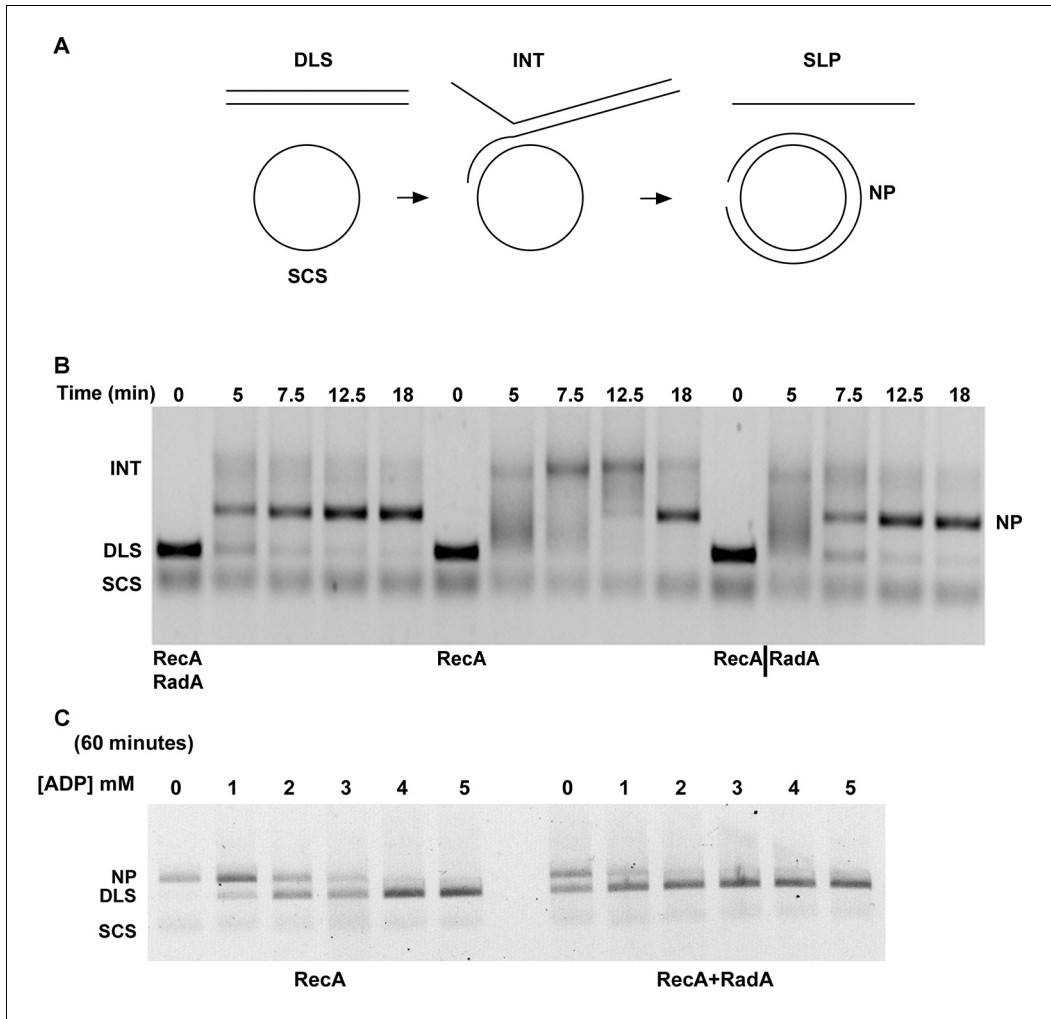

**Figure 4.** Three-strand Recombination Reactions in the Presence of RadA. (**A**) Diagram of thethree-strand recombination reaction. Single-strand circular φX-174 DNA (SCS) was mixed with double-strand φX174 DNA linearized with *Pst*I (DLS) in the presence of RecA, SSB, ATP and an ATP regenerating system. When RadA was included in the reactions, it was added to achieve a RadA:RecA ratio of 1:17. Initially, branched Intermediates (INT) form between the singe-strand circular DNA and its complementary sequence. After continued incubation, nicked circular product (NP) and single-strand linear product (SLP) are formed. Note: The SLP is not usually visible. The standard order of addition for this reaction is: 1) Incubation at 37 °C for 8 min with buffer, ATP regenerating system, φX174 single-strand DNA, and RecA (and RadA when included). 2) Addition of double-strand linear φX174 and continued incubation for 5 min at 37°. 3) Addition of pre-mixed ATP and SSB to initiate the reaction. Incubation then continued for the times indicated. 4) Deproteinization of the reaction and separation of the products from substrates on a 0.8% agarose gel run in TAE. (**B**) Effect of RadA on Three-strand Recombination Reactions. Three-strand recombination reactions were performed as described above with either RecA and RadA or RecA alone in the first incubation step. In the third set of reactions, RadA was added to reactions containing RecA (split from the RecA alone reaction) five minutes after addition of SSB and ATP. (**C**) Effect of excess ADP on Three-strand Recombination Reactions. Recombination reactions were performed as described except for the following modifications. No regenerating system was included, but the ATP concentration was increased to 5 mM. ADP was added at the concentrations indicated. Finally, incubation at 37 °C was extended to 60 min.

The following figure supplements are available for figure 4:

**Figure supplement 1.** Effect of order of addition of reaction components on 3-strand recombination reactions in the presence or absence of RadA.

**Figure supplement 2.** Titration of RecA in 3-strand recombination reactions in the presence or absence of RadA.

*Figure 4 continued on next page*

*Figure 4 continued*

**Figure supplement 3.** 3-strand recombination reactions with saturating RecA concentrations and RadA concentrations as indicated.

**Figure supplement 4.** Effect of RadA alone on 3-strand recombination reactions.

the reaction (*Figure 4—figure supplement 1*) indicating that RadA does not assist nucleation or filament extension under these conditions. Under no conditions examined did we find that RadA by itself could promote strand-pairing and exchange (for example, see *Figure 4—figure supplement 4*).

To determine which properties of RecA are required for RadA-stimulation of strand exchange, we next examined strand-exchange catalyzed by RecA K72R, a mutant that can bind but not hydrolyze ATP (*Rehrauer and Kowalczykowski, 1993*; *Shan et al., 1996*). These reactions substitute dATP for ATP, which promotes higher affinity of RecA for ssDNA, and lower $Mg^{2+}$ concentrations. As reported previously, the RecA mutant readily catalyzes strand-pairing to form branched intermediates but is inefficient at branch migration to form the final relaxed circular dsDNA product. RadA did indeed accelerate branch migration in RecA K72R-mediated reactions, with final nicked circular product ('NP') visible at 30 min (*Figure 5A*). Therefore, RadA-acceleration of branch migration in RecA-promoted reactions does not require the ATPase of RecA.

In the standard RecA 3-strand reaction, we examined the ability of RadA mutant proteins to accelerate branch migration, compared to reactions in parallel with wild-type RadA or lacking RadA altogether (*Figure 5B*). Interestingly, the RadA K258A mutant appeared to arrest branch migration, with the accumulation of slowly-migrating intermediate species. Final product formation was

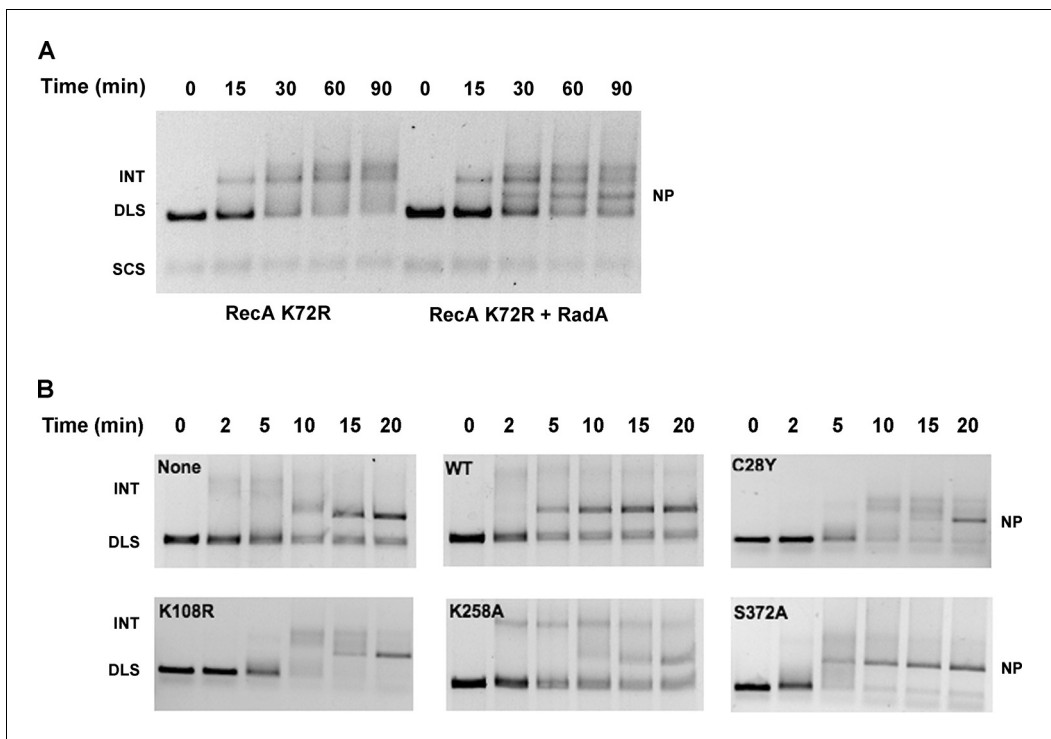

**Figure 5.** Mutational Analysis of 3-strand Recombination Reactions. (**A**) Three-strand recombination reactions with RecA K72R and RadA. Recombination reactions with the mutant RecA K72R were performed as described except dATP replaced ATP and the $Mg(OAc)_2$ concentration was decreased to 3 mM. (**B**) Three-strand recombination reactions with RadA domain mutants. The reactions were performed as described except with the indicated RadA mutant protein replacing wild-type RadA.

reduced, more so than reactions that lack RadA. RadA mutants C28Y and K108R also slowed the reactions and reduced the efficiency of strand exchange. In contrast, RadA S372A appeared fully wild-type in its ability to accelerate strand-exchange. These experiments show that RadA's ATPase activity, negated in the K108R mutant, is required to stimulate branch migration. RadA K258A, a dominant-negative mutant in the 'RadA motif', not only did not accelerate branch migration but also interfered with RecA's ability to branch migrate, suggesting it binds either to intermediate structures and/or RecA to block the reaction. RadA's putative zinc finger, affected by the C28Y mutant, is also required to stimulate branch migration.

To test whether RadA could promote branch migration in the absence of RecA, we performed standard RecA strand-exchange reactions for 12 min and purified the DNA from such reactions after proteinase K/SDS treatment to remove RecA. This DNA, enriched in branched strand-exchange intermediates, was then incubated with RadA alone in the presence of ATP. RadA caused loss of branched DNA and enhanced formation of relaxed circular dsDNA product, relative to the same substrate incubated without RadA (*Figure 6A*, quantitated in *Figure 6B*). Interestingly, branch migration catalyzed by RadA under these conditions was directional in nature, with accumulation primarily of full strand exchange products (nicked circular dsDNA product, 'NP') rather than linear dsDNA ('DLS') (*Figure 6A,B,D*). In the presence of SSB, however, the directionality of RadA-mediated branch migration was lost, with accumulation of both linear and circular products (*Figure 6A,C, D*). The ability of RadA to catalyze branch migration in this assay required ATP (*Figure 6E*) and was not seen in reactions containing no nucleotide, ADP or ATPγS.

The prior strand-exchange reactions involve three DNA strands, with one substrate and one product entirely single-stranded. RecA can also catalyze strand-exchange between two duplex molecules, provided that strand exchange is initiated at a short ssDNA gap in the substrate. These reactions produce full 4-strand Holliday junction intermediates, as opposed to the 3-strand junctions in the 3-strand reactions above (*Figure 7A*). To determine if RadA could accelerate branch migration between 4 DNA strands involving true Holliday junctions, we performed RecA strand exchange reactions between linear dsDNA molecules and circular dsDNA with a 1346 nt ssDNA gap. All other conditions were identical to the 3-strand reactions above and the 3-strand reactions were performed in parallel to the 4-strand reactions (*Figure 7B*). Although RecA by itself efficiently promoted joint molecule formation in the 4-strand reaction, formation of the final branch-migrated product (nicked dsDNA circle, 'NP') was inefficient relative to the 3-strand reaction after 30 min, and most DNA was found in various branched intermediate forms. In the RecA RadA-coupled reaction, the final product was visible at the first time point, 10 min, and accumulation of intermediates was not observed. Therefore, RadA can stimulate branch migration of 4-strand Holliday junctions, as well as 3-strand junctions.

## Discussion

RadA is a ubiquitous RecA-paralog protein found in eubacteria and plants. Genetic studies have implicated RadA in homologous recombination, particularly in the late steps of recombination intermediate processing. This work presented here provides a biochemical rationale for this role, showing that purified RadA protein can mediate branch migration of recombination intermediates, in the context of a RecA filament and in a direction 5' to 3' with respect to the initiating single-strand.

In theory, branch migration can promote homologous recombination in several ways (*Figure 8A*). If strand exchange is initiated at a site removed from a 3' end, branch migration can serve to engage the 3' strand into the heteroduplex region, providing a paired 3' end that can be extended by DNA polymerases. RadA's directionality is consistent with this role. Furthermore, RadA's inhibition by SSB may prevent the reverse reaction, the dissolution of this heteroduplex. Branch migration also can extend the heteroduplex region formed between donor and recipient DNA strands, which may aid its stability. Therefore, we might expect RadA to aid the process known as 'break-induced replication' (*Anand et al., 2013*), during which a resected linear DNA fragment invades a homologous duplex region, and establishes a replication fork. Indeed, RadA is strongly required for exchange events believed to be associated with breakage of the replication fork in vivo (*Lovett, 2006*).

Branch migration also can dissociate recombination intermediates and is integral to a recombination reaction known as 'synthesis-dependent strand-annealing' (SDSA), a process that can heal double-strand breaks (DSBs) without crossing-over (*Figure 7B*). This mechanism underlies a number of

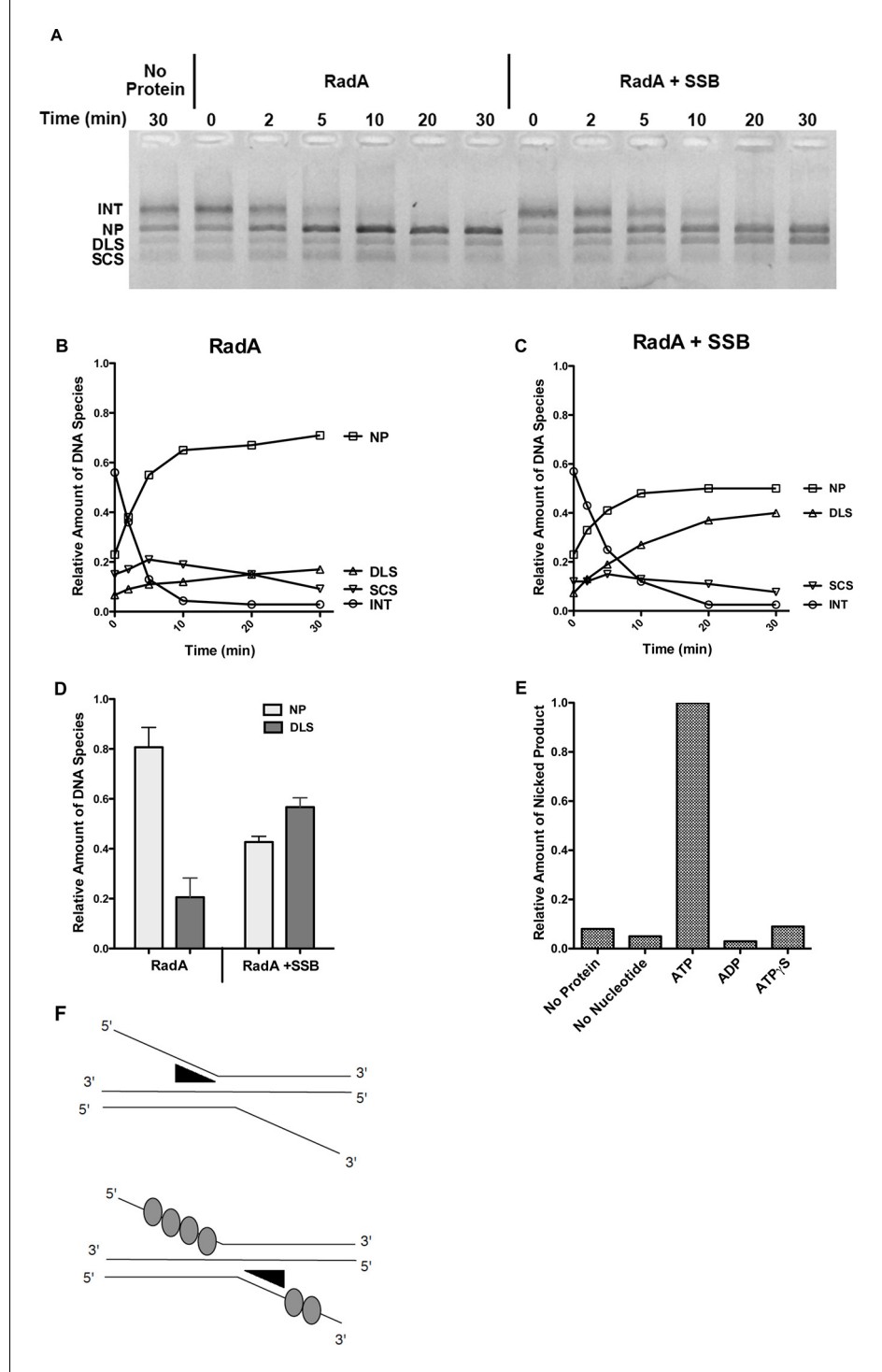

**Figure 6.** Recombination Intermediate Branch Migration. (**A**) Branch Migration of intermediates Mediated by RadA. Three-strand recombination reactions were stopped after 12 min and deproteinized and purified as described in the procedures. Branch migration assays contained DNA intermediates (100 ng), 1 μM RadA, 3 mM ATP, and 2.4 μM SSB when indicated. After incubation for the times incubated, reactions were stopped and products were resolved on an 0.8% TAE agarose gel. The No protein sample includes DNA intermediate fractions and ATP and was incubated for 30 min at 37 °C without RadA or SSB. (**B**) Quantification of DNA Species in the Branch Migration Assay Formed by RadA. Amounts of each DNA species was determined from scanned digital photographs using ImageJ64 (Nicked Product (NP)-squares, Duplex Linear Substrate (DLS)-triangles, Single-strand Circular Substrates (SCS)-inverted triangles, and Intermediate Substrates (INT)-circles). (**C**) Quantification of DNA

*Figure 6 continued on next page*

*Figure 6 continued*

Species in the Branch Migration Assay Formed by RadA and SSB.* Amounts of each DNA species was determined from scanned digital photographs using ImageJ64. (Nicked Product (NP)-squares, Duplex Linear Substrate (DLS)-triangles, Single-strand Circular Substrates (SCS)-inverted triangles, and Intermediate Substrates (INT)-circles). (D) Quantification of the Nicked Product (NP) and Duplex Linear Substrate (DLS) Formed in Branch Migration Assays. Graph shows the mean and standard deviation of the relative amounts of NP and DLS formed in three independent branch migration experiments. Two different RadA preparations and three different DNA Intermediate preparations were used in these experiments. (E) Nucleotide Dependence of the Branch Migration Assay. Reactions were performed as above except 1mM of the nucleotide indicated replaced 3mM ATP. Incubation was for 30 min at 37 °C. (F) Model Depicting RadA Directionality. In the absence of SSB, RadA (illustrated by wedge shape) preferentially migrates DNA, displacing a 5' ssDNA flap. In the presence of SSB, the directional bias of RadA branch migration is largely eliminated. * No correction for the difference in binding affinity of ethidium bromide for single-strand and double-strand DNA was made. Thus, the absolute amount of the DNA species containing single-strand DNA may be underestimated.

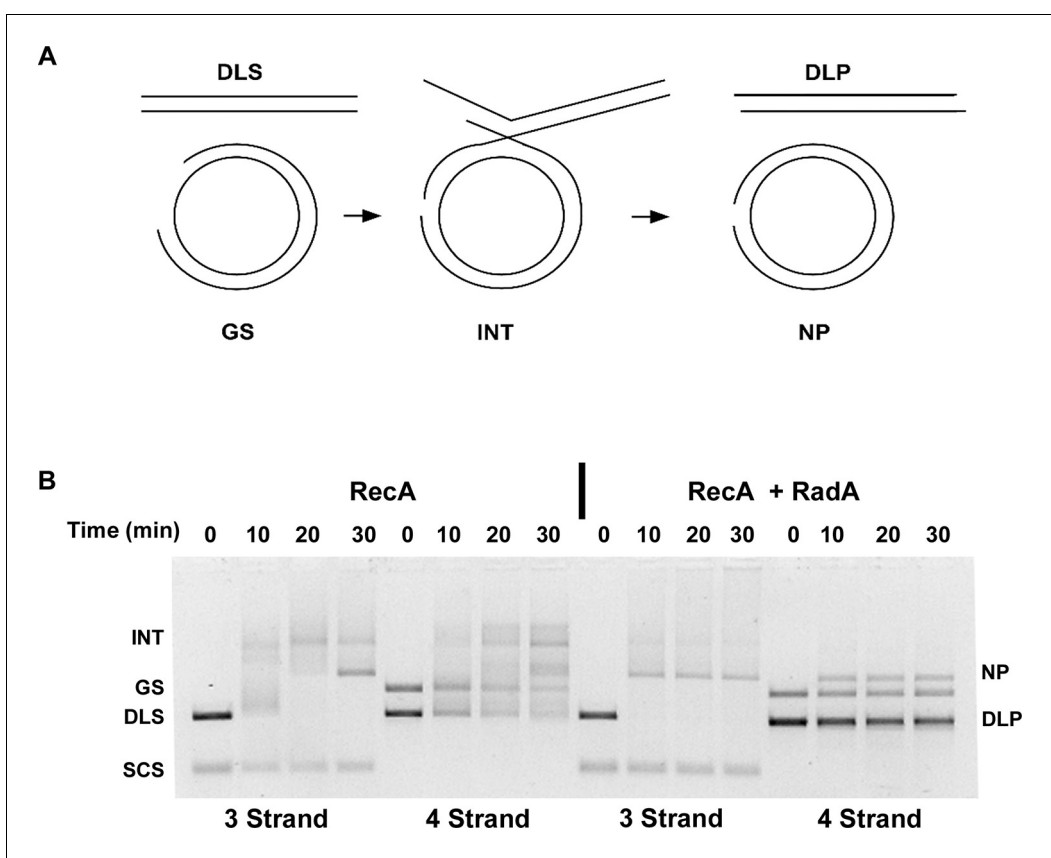

**Figure 7.** Four-strand Recombination Reactions in the Presence of RadA. (A) Diagram of the Four-strand Recombination Reaction. Gapped circular substrate (GS) prepared as described in the procedures was mixed with double-strand φX174 DNA linearized with *Pst*I (DLS) in the presence of RecA, SSB, RadA, ATP and an ATP regenerating system. Complex, largely duplex DNA intermediates are formed first. The final products are nicked circular double-DNA (NP) and Duplex Linear DNA with Single-strand Tails (DLP). Note: The tailed linear product species is not well-resolved from the duplex linear substrate (DLS). (B) Comparison of 3-strand and 4-strand Recombination Mediated by RecA in the Presence and Absence of RadA. Recombination reactions between either single-strand circular φX174 DNA (SCS) and double-strand φX174 DNA linearized with *Pst*I (DLS)-3-strand reactions or double-strand circular φX174 with a 1.3 kB single-strand gap (GS) and double-strand φX174 DNA linearized with *Pst*I (DLS)-4-strand reactions were performed as described. At the times indicated, reactions were stopped and de-proteinated. Products were resolved using an 1.0% agarose gel in TAE buffer.

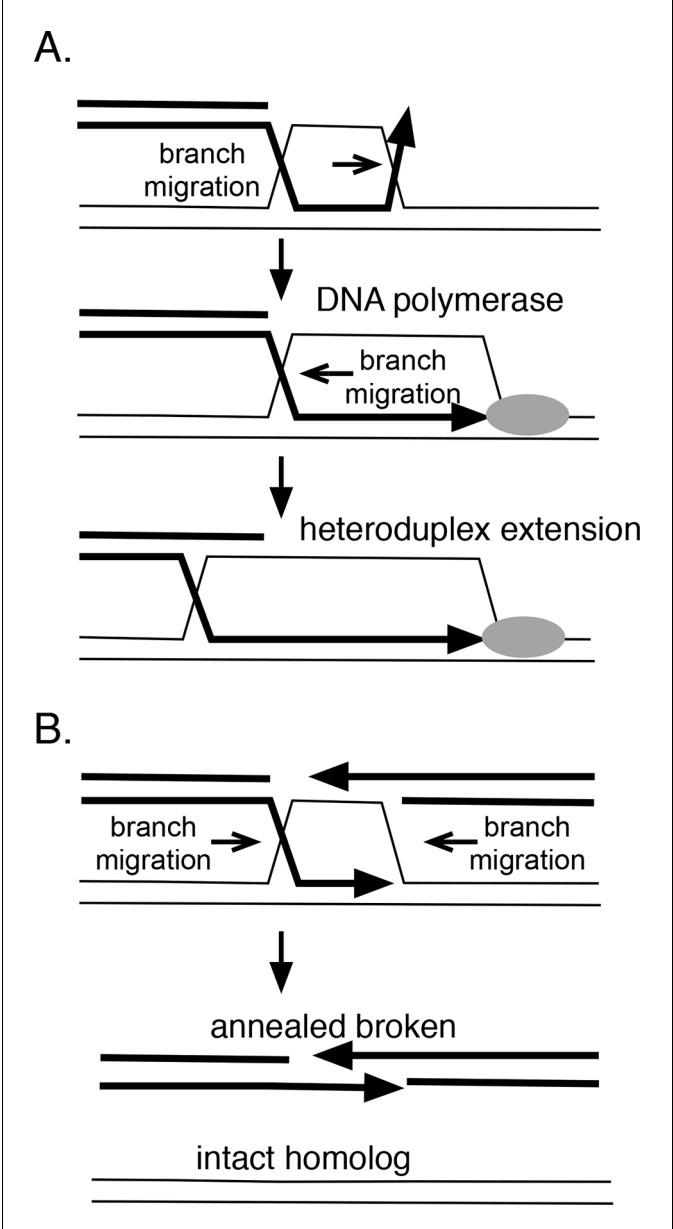

**Figure 8.** How branch migration assists homologous recombination. (**A**) Heteroduplex extension. In reactions between linear resected DNA and an intact chromosome, initial strand pairing and invasion may occur at a distance from the 3' end. Branch migration of the D-loop (in direction of the arrow) allows the heteroduplex region to extend fully to the 3' end, allowing it to be engaged by DNA polymerases. Branch migration also allows the D-loop to be extended, lengthening and stabilizing the region of heteroduplex and forming a 4-strand Holliday junction. (**B**) Synthesis-dependent strand annealing (SDSA). After resection of a broken chromosome and strand invasion into a sister molecule, branch migration is required to dissolve the intermediate, allowing broken strands to anneal to one another and the break to be healed. Reactions contained 1 mM ATP and 20.3 µM (nucleotide) φX174 DNA. The values shown are the average of two experiments, except for the circular single-strand value. It is the average of five experiments. Standard deviations are reported.

repair events associated with DSBs, including transposon-mediated breaks in Drosophila (*Nassif et al., 1994*), mating-type switching in yeast (*Haber et al., 2004*) and radiation damage-repair in the bacterium *Deinococcus radiodurans* (*Zahradka et al., 2006*). Initial homologous strand-exchange allows DNA synthesis across the region of the break; subsequent branch migration

dissolves the intermediate to allow the broken strand to anneal to itself and to be joined. In *Deinococcus radiodurans, radA* is required for the SDSA reactions that aid extreme radiation resistance in this organism (*Slade et al., 2009*). In this case, RadA protein appears to promote the initial joints that prime DNA synthesis but could, in theory, also participate in the subsequent branch migration that resolves such joints to allow annealing. Alternatively, this dissolution might be catalyzed by RecG, whose polarity seems suited for this role (Whitby et al., 1993). Catalysis of branch migration can also allow bypass of barriers such as DNA lesions or regions of non-homology that are sufficient to block spontaneous thermal branch migration.

E. coli possesses three branch migration systems that participate in recombination, RadA, RecG and RuvAB. Genetic analysis indicates that these systems are both somewhat specialized and somewhat redundant (see discussion [*Cooper et al., 2015*]). How does branch migration catalyzed by RadA differ from that promoted by RecG and RuvAB? One difference is that RadA can function in the context of the RecA synaptic filament, with branches migrated in the direction of RecA-promoted strand transfer. In contrast, when purified RuvAB or RecG are added to ongoing RecA-strand transfer reactions, they decrease the recovery of full strand-exchange products, by accelerating the reverse reaction back to substrate forms (*Whitby et al., 1993*). On RecA-free intermediates, RecG and RuvAB migrate branches preferentially to substrate and product forms, respectively.

There are two mechanisms by which proteins mediate ATP-driven branch migration, the first exemplified by the *E. coli* RuvAB complex. RuvA forms a tetramer, which specifically binds Holliday junctions; RuvB acts as two hexameric complexes, flanking RuvA and encircling duplex DNA (*Parsons et al., 1995*; *Yamada et al., 2002*). RuvB acts as the motor, pumping DNA through the complex and thereby moving the position of the junction. RuvAB has classical helicase activity (*Tsaneva et al., 1993*), unwinding DNA strands, and, through the RuvA complex, special affinity for branched structures (*Parsons and West, 1993*). The magnitude of RuvAB ATPase activity depends on the DNA structures to which it is bound (*Abd Wahab et al., 2013*). The RuvAB complex interacts with a nuclease component, RuvC, coupling branch migration to junction cleavage (*West, 1997*). In *E. coli* branch migration might be limited to providing the preferred sequence (A/T TT G/C) for RuvC cleavage (*Shah et al., 1994*; *Shida et al., 1995*). RecG, the other branch-migration protein in *E. coli*, is a DNA translocase with special affinity for branched structures; it is also believed to branch migrate DNA via a motor mechanism (*Whitby and Lloyd, 1998*; *Whitby et al., 1993*).

On the other hand, RecA protein catalyzes branch migration by a distinctly different mechanism involving strand-exchange between DNA in the primary and secondary DNA-binding-sites on the RecA filament. Site II-bound DNA has been modeled with the RecA filament as a helix with same average pitch as Site-I-bound DNA (the latter visible in the crystal structure, [*Chen et al., 2008*]); this modeled Site II-bound DNA, however, has a larger radius and an even more highly extended DNA structure than Site I-bound DNA (reviewed in [*Prentiss et al., 2015*]). ATP hydrolysis is required for RecA strand transfer over extended distances (*Jain et al., 1994*); how ATP hydrolysis promotes branch migration via the RecA mechanism is not well understood. Unlike RuvAB, RecA has no special affinity for branched DNA structures nor can it act as a DNA helicase/translocase. The mechanism by which RadA branch migrates DNA is not known, although its RecA-like sequence character, lack of structure-specific binding or helicase activity might suggest a RecA-like mechanism. Further analysis should be revealing.

Although we have no direct evidence for this, the RecA-like structure of RadA suggests that it might be recruited to a RecA filamens, interacting at its natural interface. Moreover, our ADP inhibition experiments raise the possibility that RadA may destabilize the RecA filament. In vivo, RecA foci become more numerous and persistent in *radA* mutants of *E. coli* (*Massoni et al., 2012*), consistent with a role for RadA in RecA postsynaptic filament destabilization. This property may serve to provide a handoff of recombination intermediates from RecA to RadA, facilitating the completion of recombination.

Strand-exchange paralog proteins are universally found in archaea, eubacteria and eukaryotes. Humans have five such Rad51 proteins, in addition to true strand exchange proteins Rad51 and Dmc1 (reviewed in [*Gasior et al., 2001*]). These proteins are required for homologous recombination, albeit to a lesser extent, than their true strand exchange-protein counterparts. The few that have been studied biochemically appear to affect the presynaptic phase of strand exchange by enhancing formation or stability of the Rad51 filament. Rad55/Rad57 of yeast interact with Rad51 (*Johnson and Symington, 1995*) and act as Rad51-mediator proteins to allow Rad51 to overcome

inhibition by single-strand DNA-binding protein, RPA, in formation of the Rad51 presynaptic filament (*Sung, 1997*). In the archaea, *Sulfolobus tokodaii* StRad55 protein appears to play a similar role (*Sheng et al., 2008*). In addition, yeast Rad55/Rad57 have an additional role in stabilizing the Rad51 filament against dissociation by the Srs2 helicase (*Liu et al., 2011*).

The effect of bacterial RadA on late stages of recombination, evident both in vivo and in vitro, presents a new paradigm for strand-exchange paralog proteins that may be shared in other organisms. RFS-1, the sole Rad51 paralog of *C. elegans* has properties consistent with a late recombination role and a RFS-1 peptide can disrupt Rad51 filaments in vitro (*Adelman and Boulton, 2010*), although recent evidence supports a presynaptic role in remodeling the Rad51 filament to a more flexible form (*Taylor et al., 2015*). The human Rad51C-XRCC3 paralog complex may also have a late recombination role: depletion of Rad51C from cell extracts reduces branch migration capacity and copurifies with HJ cleavage activity (*Liu et al., 2004*; *2006*).

Although RecA can promote the homology search process, pair DNA and promote branch migration, RadA may be a specialized form, selected for its ability to catalyze faster branch migration and incompetent for homology-search and pairing. Because of its role in synapsis, RecA binding needs to be highly specific for ssDNA, lest it bind indiscriminately to the undamaged chromosome. RadA cannot pair DNA and has weak ability to bind ssDNA, in comparison to RecA. Its capacity to bind DNA in the presence of ADP is an intriguing property. In a study of a RecA mutant (P67G E68A, near the Walker A motif), strand-exchange between lengthy homologies (but not homologies <2 kb) was highly stimulated by ADP and completely inhibited by ATP or an ATP-regenerating system. This finding suggests that the ADP-form of the RecA filament is required, in some way, for the branch migration phase of strand-exchange, as revealed by this particular mutant. RadA binding behavior may naturally reflect this propensity and the stable binding of RadA in the presence of ADP may explain its superior ability to promote branch migration, relative to that by RecA. RadA has stronger dsDNA-stimulated ATPase activity relative to RecA, which may also assist branch migration. Because the role of ATP hydrolysis in RecA strand-exchange is still unclear, further study of RadA-mediated branch migration may provide valuable insights into this mechanism.

Our study shows that RadA's ATPase, 'RadA motif' (KNRFG) and Zn-finger motif are essential to the biochemical function of the protein, consistent with our prior genetic results (*Cooper et al., 2015*). We hypothesize, based on the position of KNRFG element in the RadA sequence, that this motif is required to assemble the ATPase site (comparable to K248 K250 region of RecA [*Chen et al., 2008*]) at an interface. The Zn-finger appears to assist ssDNA binding but it may also facilitate some protein (SSB, RecA?) or DNA interaction (branched molecules?), since it is required for RadA's stimulation of branch migration in RecA-coupled reactions.

Our study does not address the ability of RadA to form a filament or other oligomeric structure with itself or with RecA, an area for further investigation. Threading of RadA onto the RecA presynaptic crystal structure (our unpublished results) suggests that RadA possesses subunit interfaces similar to that of RecA that assemble the ATPase site. In a bacterial one-hybrid assay, RadA was shown to interact with itself (*Marino-Ramirez et al., 2004*), indicating that it forms a multimeric complex. Our early experiments with His$_6$-tagged RadA protein, a less active protein than the more native protein characterized here, exhibited multiple bound species in gel-shift experiments with poly(dT) (data not shown), consistent with oligomer formation. RadA's ability to reduce ATPase activity of RecA in the presence of ssDNA and SSB, its enhancement of ADP-inhibition of RecA-mediated strand exchange and the ability of RadA K258A (and to a lesser extent C28Y and K108R) to inhibit RecA-mediated branch migration are consistent with the notion that RadA joins and destabilizes the RecA filament. Biochemical confirmation of this hypothesis is ongoing.

## Materials and methods

### Materials

Biochemicals were purchased from USB or Sigma unless noted. Wild-type RecA was a kind gift from Shelley Lusetti (New Mexico State University) or purchased from Epicentre Biotechnologies (Madison, WI). D72R Mutant RecA was generously provided by Michael Cox (University of Wisconsin-Madison). Single-strand DNA-binding Protein (SSB) was from Promega (Madison, WI) and T4 polynucleotide kinase and restriction enzymes were from New

**Table 2.** Oligonucleotides used in this study.

| Oligonucleotide name | Sequence |
|---|---|
| radAeXactF | GGAAGCTTTGACTTCTGTGGCAAAAGCTCCAAAACG |
| radAeXactR | TTTGCGGCCGCTTATAAGTCGTCGAACACGC |
| polyd(N)$_{33}$ | TTAGCGGCCGCATAGTCAAGATGACAATGTTCT |
| Substrate E2 | CGGTCAACGTGGGCATACAACGTGGCACTG (T)$_{30}$ATGTCCTAGCAAAGCGTATGTGATCACTGG |
| Jxn1 | CCGCTACCAGTGATCACCAATGGATTGCTAGGACATCTTTGCCCACCTGCAGGTTCACCC |
| Jxn2 | TGGGTGAACCTGCAGGTGGGCAAAGATGTCCTAGCAATCCATTGTCTATGACGTCAAGCT |
| Jxn3 | AGCTTGACGTCATA |
| Jxn4 | GATCACTGGTAGCGG |
| Jxn5 | TGCCGATATTGACAAGACGGCAAAGATGTCCTAGCAATCCATTGGTGATCACTGGTAGCGG |

England Biolabs (Ipswich, MA). Regenerating system enzymes were from Sigma-Aldrich (St. Louis, MO). φX-174 and M13 RF and virion DNAs were purchased from NEB, pBS SK-was from Agilent Technologies (Santa Clara, CA) and pPAL7 was from Bio-Rad Laboratories (Hercules, CA). Oligonucleotides were purchased from Sigma-Aldrich.

## Cloning of radA and site-directed mutagenesis

The wild-type RadA gene was amplified by PCR and cloned into the high copy vector pBS SK-. Site-directed mutants were made from this construct by PCR as described (*Cooper et al., 2015*). Subsequently for protein purification, wild type and mutant RadA DNA was amplified from the pBS SK- constructs using the eXact primers (*Table 2*) and then cloned into the vector pPAL7. All constructs were confirmed by sequencing.

## RadA protein purification

Wild-type or mutant eXact-tag constructs were transformed into BL21 Codon-Plus or BL21 AI (Agilent Technologies) deleted for *endA*::Km. Typically, the transformation mix was grown for 1 hr at 37 °C in SOC media (2% tryptone, 0.5% yeast extract, 20 mM glucose, 10 mM NaCl, 10 mM MgCl$_2$, 2.5 mM KCl), diluted into 20 ml SOC supplemented with ampicillin ('Ap', 100 µg/ml) and then grown standing overnight. This culture was used as the inoculum for a 1 l LB (2% tryptone, 1% yeast extract, 0.5% NaCl) culture with Ap. Cultures were grown until the A$_{590}$ reached approximately 0.8 at which time IPTG (Gold Biotechnology, St. Louis, MO) was added to 1 mM and arabinose was added to 0.2% (for BL21 AI strains). Growth was continued for 3–4 hr at 30 °C. For production of RadAC28Y and RadAK108A mutant protein, growth conditions were altered so that a 500 ml culture was grown from the initial 20 ml inoculum in LB supplemented with 0.2% glucose and 100 µg/ml ampicillin until the culture reached an A$_{590}$ of 1.0. Cells were then diluted into LB with 0.4% arabinose, 1 mM IPTG and fresh ampicillin. Growth was continued 2 hr at 30 °C. After growth of all strains, cells were collected by centrifugation, and the resulting pellet was frozen and stored at -20 °C. RadA was purified with slight modifications from the Biorad eXact protocol. Cleavage of the N-terminal eXact tag produces a RadA protein with the addition of a 2 N-terminal amino acids, threonine serine, (necessary to facilitate efficient cleavage). Typically, pellets from 200 ml of wild type RadA culture were resuspended in 20 ml eXact buffer (100 mM sodium phosphate, pH 7.2, 10% glycerol, 10 mM beta-mercaptoethanol, 300 mM sodium acetate. (The final pH was adjusted to 7.2 if necessary). Cells were lysed by adding lysozyme (in eXact buffer) to 200 µg/ml and incubating for 45 min on ice. Lysis was completed by incubation at 37 °C for 2 min followed by homogenization using a Dounce Homogenizer on ice (5 passes with B, followed by 5 passes with A). The crude lysate was clarified by centrifugation at 17000 x g for 30 min at 4 °C. One third of the cleared lysate was applied to a 1 ml eXact pre-packed column equilibrated in eXact buffer at room temperature. The column was then washed with 15–20 ml eXact buffer. To cleave the eXact Tag from RadA, 2 ml cleavage buffer (eXact buffer + 100 mM NaF) was applied to the column, and the column was capped and incubated at room temperature for 45–60 min. RadA was then

eluted from the column with eXact buffer + 100 mM NaF. Fractions containing RadA were pooled, diluted in small batches in Q buffer (20 mM Tris-HCl, pH 8.0, 0.5 mM EDTA, 15% (v/v) ethylene glycol, 10 mM beta-mercaptoethanol) without salt to give a conductivity of approximately 125 mS$^{-1}$ and applied to a 1 ml Q HP column (GE Healthcare) equilibrated with125 mM NaCl Q buffer. After washing with 15 ml 125 mM NaCl Q Buffer, RadA was eluted with a linear gradient from 125 mM NaCl in Q buffer to 750 mM NaCl in Q buffer with 20% glycerol replacing the ethylene glycol. RadA eluted at a conductivity approximately 275 mS$^{-1}$. To retain activity, fractions containing RadA were immediately flash-frozen in small aliquots and stored at -80 °C. Thawed aliquots were used within 24 hr. Purification from 200 ml of cells yielded approximately 0.75–1 mg of highly purified RadA Protein (*Figure 1—figure supplement 1*). For some experiments, purified RadA was concentrated using Q HP Sepharose beads (GE Healthcare, Chicago, IL). RadA protein concentration was spectrophotometrically determined using an extinction coefficient of 22,460 M$^{-1}$ cm$^{-1}$ (*Gasteiger et al., 2005*) or by the Bradford method (*Bradford, 1976*), which gave equivalent results.

## ATPase assays

ATP hydrolysis activity of RadA mutants was tested either by measuring release of inorganic $^{32}$P from ATP (PerkinElmer, Waltham, MA) using thin layer chromatography (TLC) with polyethylenimine plates (PEI, Macherey-Nagel, Düren, Germany) as the solid phase and 0.5 M LiCl/ 4.3% formic acid as the mobile phase (*Kornberg et al., 1978*). Reactions were incubated at 37 °C for the times indicated and contained 10 mM Bis-Tris-Propane-HCl, pH 7.0, 10 mM MgCl$_2$, 2 mM DTT, 1mM ATP. ATP hydrolysis by wild-type RadA protein was measured in reactions that included an ATP regenerating system and were coupled with NADH oxidation. Oxidation of NADH was monitored spectroscopically at 380 nm (Extinction coefficient = 12100 M$^{-1}$cm$^{-1}$) using a Synergy H1 Microplate Reader and Gen5 Data Collection and Analysis Software (Biotek, Winooski, VT). Reactions included 1 µg of φX174 single-strand circular DNA, 10 mM Bis-Tris-Propane-HCl, pH 7.0, 10 mM MgCl$_2$, 2 mM DTT, 1mM ATP, 2 mM DTT, 3.5 mM phosphoenol pyruvate, 10 u/ml pyruvate kinase, 2 mM NADH, and 10 u/ml lactate dehydrogenase. For all reactions, ATP concentrations were determined at 260 nm using an extinction coefficient of 15400 M$^{-1}$cm$^{-1}$

## DNA-binding experiments

Oligonucleotides (*Table 2*) were 5' end-labeled with $^{32}$P-ATP using T4 polynucleotide kinase and manufacturer's conditions (New England Biolabs). Excess $^{32}$P-ATP was removed from the reaction using Sephadex G-50 columns (Roche, Basel, Switzerland). Double-strand substrates were made by heating two complementary oligonucleotides to 95 °C and then cooling to room temperature slowly. Standard DNA-binding reactions contained RadA as indicated, 100 fmol DNA, 50 mM Tris-HCl buffer, pH 7.5, 10 mM MgCl$_2$, 0.1 mM EDTA, 75 mM NaCl, 5 mM dithiothreitol, 100 µg/ml bovine serum albumin and 1 mM nucleotide. Reactions were incubated at 37 °C for 30 min and then resolved on a 6% Tris-borate EDTA polyacrylamide gel (pre-run for 1 hr) at 100 V for 45 min at room temperature. Gels were then dried and binding was analyzed using ImageJ-64 software with scanned autoradiographs (HiBlot CL film-Denville Scientific, Holliston, MA).

## Model-branched substrate formation

To form branched substrates, equimolar concentrations of oligonucleotides (*Table 2*) were mixed and heated to 100° for 5 min followed by slow cooling to room temperature in 10 mM Tris Acetate, pH 7.4, 10 mM Mg Acetate, and 50 mM K Acetate. The extent of branched molecules formation was assessed either by electrophoresis using 3% agarose gels in TAE followed by staining with ethidium bromide or by using oligo 2 radio-labeled with polynucleotide kinase and $^{32}$P-ATP to form branched molecules (in parallel reactions) followed by electrophoresis on 6% acrylamide gels in TBE and autoradiography. Contaminating structures were present at less than 5% of the total, except for the 3-strand fork when contaminating structures were between 5 and 10%. Annealed substrates were used without further purification. **Fork 1** was made by annealing Jxn1 and Jxn 2 oligonucleotides. **Fork 2** was made from Jxn1, Jxn 2, and Jxn3 oligonucleotides. **Fork 3** was made by annealing Jxn1, Jxn2, and Jxn4 oligonucleotides. **Fork 4** was the annealed product of Jxn1, Jxn2, Jxn 3, and Jxn 4 oligonucleotides. The **3-stranded fork** was constructed from Jxn1, Jxn2, and Jxn 5. oligonucleotides. *Figure 2—figure supplement 1* shows a diagram of each branched substrate structure.

## Strand exchange reactions

### Three-strand reactions

Recombination between single-strand circular φX174 virion DNA and double-strand RF φX174 linearized with *Pst*I (New England Biolabs) were performed as followed with exceptions noted in the figure legends. Reactions contained 20 mM Tris-acetate pH 7.4, 12.5 mM phosphocreatine, 10 u/ml creatine kinase, 3 mM ammonium glutamate, 1 mM dithiothreitol, 2% glycerol, 11 mM magnesium acetate. Under standard conditions, 6.7 µM RecA and RadA as indicated (most typically 400 nM) was incubated with 20.1 µM (in nucleotide) viral φX174 DNA for 8 min at 37 °C. Then, 20.1 µM linear double-strand φX174 DNA was added and incubation was continued for 5 min. The reactions were initiated by the addition of ATP to 3 mM and SSB to 2.1 µM. After incubation for the times indicated, reactions were quenched by addition of EDTA to 15 mM and SDS to 1.25%. Recombination products were separated on an 0.8% Tris-acetate EDTA (TAE) agarose gel run at 5.5 V/cm and then visualized by staining with ethidium bromide.

### Four-strand reactions

Reactions between a circular duplex φX174 DNA with a 1346 base single strand gap and duplex linear DNA were performed as for 3-strand reactions except products were analyzed on a 1% agarose (Biorad) gel in TAE. The *Pst*I linear fragment shares homology with 836 bases on one end and 516 bases on the other end with the single-strand region of the gapped molecule. Product formation was quantified using ImageJ-64 software.

## Gapped DNA molecule formation

φX174 double strand circular DNA was cleaved with *BsaI* and the 4.1 kb fragment was purified from 0.7% low-melt agarose (USB, Cleveland, OH) using Gene Jet gel purification columns (Thermo-Fisher Scientific, Waltham, MA). Gapped molecules were made using a large-scale recombination reaction using conditions above (without RadA) with the *BsaI* fragment replacing the full length linear fragment and incubation extended to 1.5–2.0 hr. The reaction mix was then extracted with phenol two times. After back-extracting the organic phase with 1 volume of 10 mM Tris-EDTA, all aqueous phases were combined and extracted with chloroform:isoamyl alcohol (24:1). Sodium acetate, pH 5.2 was added to 0.3 M and DNA was precipitated using 2.5 volumes of ethanol. DNA pellets were then washed with 70% ethanol and resuspended in 10 mM Tris-EDTA. Finally, gapped DNA was gel-purified as described above.

## Recombination intermediate isolation and branch migration assays

Large-scale standard recombination reactions were stopped with after 12 min by addition of EDTA to 35 mM, SDS to 0.65%, and Proteinase K to 800 µg/ml and applied to a 3.5 ml Sepharose 4B-CL equilibrated in 20 mM Tris-acetate pH 7.4, 3 mM ammonium glutamate, 1 mM DTT, 2% glycerol, and 11 mM magnesium acetate. Fractions (150–200 µl) containing DNA were identified by staining with Picogreen.

## Branch migration assay

DNA fractions containing recombination intermediates were incubated at 37 °C in 20 mM Tris-acetate pH 7.4, 3 mM ammonium glutamate, 1 mM DTT, 2% glycerol, 11 mM magnesium acetate, and approximately 30 mM NaCl from the RadA protein with RadA (460 nM), 3 mM ATP, and, when included, SSB (2.1 µM). Reactions were stopped and analyzed as outlined above.

## Acknowledgements

This work was supported by NIH grant R01 GM51753. We thank Shelley Lusetti (University of New Mexico Las Cruces) and Michael Cox (University of Wisconsin) for provision of RecA protein.

## Additional information

### Funding

| Funder | Grant reference number | Author |
|---|---|---|
| National Institute of General Medical Sciences | GM51753 | Susan T Lovett |

The funders had no role in study design, data collection and interpretation, or the decision to submit the work for publication.

### Author contributions

DLC, Conception and design, Acquisition of data, Analysis and interpretation of data, Drafting or revising the article; STL, Conception and design, Analysis and interpretation of data, Drafting or revising the article

### Author ORCIDs

Susan T Lovett, http://orcid.org/0000-0003-2792-1857

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
