## [Decision Letter]

Thank you for submitting your work entitled "Recombinational branch migration by the RadA/Sms paralog of RecA in *Escherichia coli*" for peer review at *eLife*. Your submission has been evaluated by three expert reviewers, one of whom is a member of our Board of Reviewing Editors. All of the reviewers found your work to be interesting and believe that it has the potential to be published in *eLife*. However, they also found the work to be too preliminary for publication in its present form, and significant changes to the manuscript will be required before it can be considered further.

As you will see from the detailed reviews below, the reviewers ask for the following:

The DNA binding studies are incomplete and need further experimentation. In particular, the binding of RadA to branched structures needs to be carried out in order to confirm that RadA is true branch migration protein as suggested by the key data in the manuscript. The present data showing that RadA only binds poly(dT), while many other DNAs stimulate its ATPase activity, makes little sense and needs to be developed further.

From the branch migration data presented in the paper it is to be expected that RadA (like RuvAB and RecG) will demonstrate DNA helicase/translocase activity. These experiments need to be carried out. These experiments can also be used to clarify the polarity issues raised by the reviewers.

All three reviewers have extensive and specific comments that should be addressed. While further experimentation will take some time to complete, I am sure that it will be worthwhile and lead to significant improvements to the manuscript.

*Reviewer #1:*

This manuscript describes the biochemical properties of the *E. coli* RadA protein. While potentially interesting, the manuscript disappoints due to the poor (or incomplete) quality of some of the datasets. The interpretation of several experiments is also questionable. Some examples:

DNA binding and ATPase assays: The work suggests that RadA binds to poly(dT) but not poly(dA), (dC) or (dG). Yet these (non-bound) polynucleotides stimulate the ATPase activity of RadA indicating that they do in fact interact with RadA. Presumably the interaction with poly(dT), observed by PAGE is a reflection of the stability of binding rather than binding per se – especially since binding is only seen in the presence of ADP. The turnover rates on the other substrates (and with ATP) may be significantly greater than that observed with poly(dT). Have attempts been made to stabilize complexes by glutaraldehyde fixation? Also, from the gel data provided it is not clear whether the observed binding band represents a defined RadA-DNA complex or an aggregate that fails to enter the gel. Binding assays would benefit from the use of quantitative assays such as fluorescence anisotropy or surface plasmon resonance, rather than gel assays. Given that RadA appears to drive junction branch migration, the assays should be carried out with single-stranded (both defined and random sequence), duplex, 3-arm and 4-arm (Holliday junction) DNA substrates. Only then will the substrate specificity become apparent. Overall, the sections on DNA binding and ATPase activity are naïve and let the manuscript down.

Branch migration assays: Some of the data showing that RadA promotes branch migration is of low quality. For example, Figure 3 looks completely different from the other figure panels, and I find it difficult to conclude anything from the data of Figure 5.

The 4-strand reactions shown in Figure 6 are also problematic, since there appears to be a substantial amount of contaminating single-stranded DNA (SCS) present. Or is this experiment a comparison of the 3- and 4-strand reaction (the legend of the time course does not say what the difference is between the first 4 and the second 4 lanes). As a reader I can only guess. Also, interpretation of these experiments, and the 3-strand reaction described earlier would greatly benefit from the use of 32P-labels. Incidentally, what is a 1.2 gap (as stated in legend).

Other enzymes that promote branch migration, such as RuvAB and RecG, are DNA translocases/helicases. To make the argument that RadA is a true branch migration protein this needs to be demonstrated for RadA.

Polarity: The directionality of RecG is 3'-5' relative to single stranded DNA, whereas RuvAB is 5'-3' (the same as RecA). These helicase polarities translate as directionality in the branch migration reactions they promote (at least with 3-stranded reactions). The reader is led to assume that RadA is a translocase similar to RecG or RuvAB – if so then the authors should show it and determine the polarity (rather than propose – without any evidence – that there is a polarity in the way that RadA is recruited to RecA filaments).

Summary: While the manuscript has some aspects that are potentially interesting, much of the data provided is not of the standard expected for publication in *eLife*. Also, essential experiments to convince the reader that this is a true branch migration protein are missing – the authors need to demonstrate that RadA shows specific binding to DNA substrates containing 3- or 4-way junctions, and to show that the protein is a DNA translocase (helicase). Without such data the work is simply too preliminary for publication in *eLife*.

*Reviewer #2:*

The bacterial RecA protein plays an important role in the reconstitution of collapsed replication forks, particularly those in which one arm of the fork has been detached as a double strand break. The reactions promoted by RecA can carry out the key reactions needed to reconstruct the fork. However, there has always been a problem. RecA-promoted DNA strand exchange has been too slow to account for the rates of fork reactivation observed in vivo.

In this important work, Dr. Lovett and colleagues provide the first in vitro characterization of the *E. coli* RadA protein, a protein that likely solves the strand exchange rate problem. The authors convincingly demonstrate that the addition of RadA to RecA reactions provides an impressive boost to the rates of many RecA reactions, complementing RecA in these endeavors. Although the work opens many new questions about how the rate enhancements are brought about, the overall effort is an extremely important step and provides a window into the (until recently) obscure bacterial RadA protein that should stimulate much additional work. A series of thoughtfully designed point mutants also reveal that the key RadA activities are dependent on the ATP hydrolysis, Zn finger, and unique KNRFG motifs of RadA. While these data provide a great addition to our knowledge of homologous recombination, the following issues need to be addressed:

The only RadA concentrations used in this study when RecA is present appear to be those needed to effect a 1:17 ratio with RecA (subsection “Strand exchange reactions”, first paragraph; Figure 4 legend). The reasons for this ratio, or results from other concentrations of RadA, are never presented. Some information should be provided as to why this ratio was chosen and what happens if other RadA concentrations are used. Are the effects of RadA saturated at these concentrations? Can lower levels of RadA suffice? Such information might provide a clue as to how processive or distributive the RadA function is.

The authors appropriately highlight the fact that RadA migrates the junction in the same direction as RecA (Abstract and end of Introduction). However, the presence of SSB, which will be ubiquitous in the cell, reverses this directionality in experiments where RecA is removed prior to RadA addition (Figure 5). The implications and context of this change in directionality are not discussed.

What is the difference between the DNA substrates in Figure 4 and those in Figure 4—figure supplement 1? The absence of 0 timepoints in Figure 4—figure supplement 1 makes it hard to figure out what is what.

In Figure 4, perhaps the lower yield of nicked DNA products when RadA is used with RecA K72R is worth commenting on.

*Reviewer #3:*

Genetic evidence supports role for RadA in *E. coli* in the late steps in recombination, potentially involving branch migration of DNA recombination intermediates. In this manuscript, the authors purified wild type RadA and several mutant proteins, and showed that RadA increases the rate of RecA-mediated recombination in vitro by stimulation of branch migration. Authors also showed that RadA preferentially binds single-strand DNA in the presence of ADP and that it exhibits ATPase activity stimulated by DNA. The results are interesting and novel, and given that *E. coli* recombination has been very extensively studied, the findings are particularly unexpected. The general finding that RadA can convert 3-stranded joint molecule intermediates into Holliday junctions is important.

That said, the manuscript is not without its flaws. The presentation of some data in some figures is not well done, and has the feeling more of a lab report, rather than manuscript written for a broad readership. Finally, putting editorial and presentation issues aside, the results presented do support the authors' conclusions. However, the choice of some of the experiments that were done is at best quirky, and in some cases, stopping just short of what could have/should have been done.

Knowing that RadA promotes migration of 3-way and 4-way junctions, the DNA binding and ATP hydrolysis experiments in this manuscript stop short of supporting the proposed the branch migration function of RadA during recombination: they fall into the category of basic characterization. Perhaps though not essential, the authors should have tested 4-way junctions or branched DNA substrates in their DNA binding and ATPase activity assays: they only tested ssDNA and dsDNA and consequently those data do not directly address the function of RadA during branch migration. Furthermore, in the competition experiments, they showed that RadA binds to ssDNA especially poly dT, but did not test any branched structures. Enigmatically, they showed that phiX174 ssDNA stimulated the ATPase of RadA, although RadA apparently does not bind to this DNA in mobility shift experiment ("data not shown"). These results do not help to understand the branch migration function of RadA.

The choice to present the data in Figure 2 as essentially raw data without any labels makes it nearly unintelligible. The authors should relegate this figure (corrected as indicated below) to the supplementary info, and instead provide panel A as a table or bar graph; B, as an x-y graph, and C as a table or bar graph.

The same comment applies to Figure 3: The authors should relegate this figure to the supplementary info, and instead provide panels A and B as a table or bar graph.

More specific comments:

In the Abstract, the authors say "RadA (Sms)" but there is no explanation for what "Sms" is even in the main text and its relationship to RadA.

In the Abstract it is stated: "Unlike other branch migration factors RecG and RuvAB, RadA stimulates branch migration in the direction of RecA-mediated strand exchange", and in the Discussion "In contrast, when purified RuvAB or RecG are added to RecA-strand transfer reactions, they decrease the recovery of full strand-exchange products, by accelerating the reverse reaction back to substrate forms (Whitby et al., 1993)." However, RuvAB can branch migrate in the same direction as RecA. In the paper cited by authors, (Witby et al., 1993), RuvAB was shown to stimulate RecA-mediated 4-strand exchange to produce final nicked product. Other papers also showed that RuvAB-mediated branch migration direction can be opposite to RecG or bidirectional (Matthew and Lloyd (1995) EMBO-J 14 3302-3310; Iwasaki et al. (1992) Genes Dev. 6 2214-2220; Tsaneva et al. (1992) Cell 69 1171-1180).

In the second paragraph of the Introduction: "ATP binding, but not hydrolysis, is required for RecA filament formation." This should be changed to "active filament" formation. RecA can make filaments in the absence of nucleotide, though it is compressed and inactive.

Introduction, sixth paragraph and onward: It will be very helpful for readers if a schematic of RadA, showing the three conserved domains, were provided.

Figure 1—figure supplement 1: There is no indication of the molecular weights of marker proteins. The contents of each lane are not described. The gel of lane Q does not have molecular weight marker. Also, please indicate with an arrow which bands in the gel are the relevant bands.

RadA mutant purification: The authors purified RadA wild type and C28Y (Zn finger), K108R (Walker A box), K258A (KNRFG RadA motif), and S372A (putative Lon protease active site) mutant proteins. However, they only showed wild type protein in Figure 1—figure supplement 1. They need to show all of their purified proteins that were used in this manuscript.

Figure 1—figure supplement 2. The gel in this figure needs to be quantified. A graph with the fitted Kd needs to be shown.

Figure 1—figure supplement 3 – It would be most helpful to the reader if the authors showed the DNA substrates in diagram and which DNA is labeled with 32P.

In the second paragraph of the subsection “DNA binding”: The sources and average lengths of poly(dT), poly(dA), poly(dG) and poly(dC) are not provided.

In the second paragraph of the subsection “DNA binding”: The authors concluded that the binding affinity is 90 nM. However, the molecule concentration of p(dT)_30_ was 11 nM, so the nucleotide concentration is 330 nM. Therefore, the result in Figure 1—figure supplement 2 may show that the stoichiometry between RadA and ssDNA is 1:2 (~180:330) but not that the binding affinity is 90 nM. To determine the binding affinity, one should use the less concentration of substrate than the binding affinity.

In the second paragraph of the subsection “DNA binding” and entire manuscript: "substrate E2" and "substrate E-2" are mixed in the text.

In the subsection “DNA-stimulated ATPase”: "these DNA molecules are not bound by RadA detectably in gel shift experiments (data not shown)". Were these experiments done in the presence of ATP or ADP?

Figure 2: The symbols of plots are not clear in the figure. The authors must use larger symbols and color is recommended.

Figure 4: Strand exchange reactions. The experimental protocol is confusing in the manuscript. In the subsection “Strange exchange reactions”, they stated "Presynaptic filaments are formed by incubation of RecA with ssDNA in the presence of ATP, SSB is then added and the reaction is initiated by the addition of the linear dsDNA". But then the Materials and methods "6.7 μM RecA and RadA as indicated (most typically 400 nM) was incubated with 20.1 μM viral φX174 DNA f 436 or 8 minutes at 37o. Then, 20.1 μM linear double-strand φX174 DNA was added and incubation was continued for 5 minutes. Reactions were initiated by addition of ATP to 3 mM and SSB to 2.1 μM". However, in the figure legend, "the standard procedure for the reaction is: 1) Incubation at 37 for 8 minutes of buffer, ATP regenerating system, φX174 single-strand DNA, and RecA (and RadA when included). 2) Addition of double-strand linear φX174 and continued incubation for 5 minutes. 3) Addition of ATP and SSB and incubation for the times indicated. Addition of SSB and ATP is critical in this experiment because RecA bind to ATP and RecA also competes with SSB for DNA binding during experiment." It is essential that the authors clearly and unambiguously define their experimental conditions.

Figure 4: "Lane 2-Standard order with no RadA" Why hasn't the joint molecule accumulated after 15 min? In Figure 4, the linear dsDNA band disappeared at 15 min and was completed converted to intermediate and a bit of nicked product, with RecA but without RadA.

Figure 4: "Lane 3-Stand order with RadA" -> "Lane 3-Standard order with RadA".

In the subsection “Strand exchange reactions”: "RecA-mediated strand exchange at 5 minutes" According to the figure legend, the incubation time was 15 min. Is 5 or 15 min correct?

In the subsection “Strand exchange reactions”: "we performed standard RecA strand-exchange reactions for 15 min and purified the DNA from such reactions after protease treatment". According to figure legend, the incubation time with RecA was 12 min. Is 12 or 15 min correct?

Figure 5: Has this experiment been reproduced? There are no error bars. Also, add labels to the lines, and use different colors for each data set as appropriate.

Figure 6: There are two sets of experiments for both RecA only and RecA-RadA reactions. What was the difference in reaction conditions for each half of each reaction?

In the subsection “Strand exchange reactions, sixth paragraph: "D108R" -> "K108R".

In the subsection “Strand exchange reactions”, last paragraph and in the Discussion, fourth paragraph: The experimental data supporting the 3-strand reaction in the presence of ADP is apparently not shown in the manuscript. This paragraph must be deleted, or the corresponding data needs to be presented.

Discussion, fifth paragraph: "true recombinases" -> "DNA strand exchange proteins".

Figure 6 legend: "with a 1.2 single-strand gap". What is 1.2? knt?

---

## [Author Response]

*As you will see from the detailed reviews below, the reviewers ask for the following: The DNA binding studies are incomplete and need further experimentation. In particular, the binding of RadA to branched structures needs to be carried out in order to confirm that RadA is true branch migration protein as suggested by the key data in the manuscript. The present data showing that RadA only binds poly(dT), while many other DNAs stimulate its ATPase activity, makes little sense and needs to be developed further.*

The branch migration experiment has been redone (multiple times) and more convincing data that RadA can branch migrate junctions is found in new Figure 6. We have repeated the DNA binding experiments and can show, by gel shifts, that RadA has no more affinity to several branched substrates than to ssDNA. It binds in the 1-10 µM range, rather than the 100 nM range for polydT; these concentrations under these conditions border on the solubility of the protein. The ATPase data are more convincing and is presented in Figure 2—figure supplement 1, for oligonucleotide substrates including ss, splay, fork, fork with 5’ gap, fork with 3’ gap and 3-strand Holliday junction, which support similar ATPase rates. The latter substrate mimics the substrate, which RadA can convincingly branch migrate (Figure 6). RadA obviously does not bind very stably to DNA structures that nonetheless stimulate its ATPase activity.

*From the branch migration data presented in the paper it is to be expected that RadA (like RuvAB and RecG) will demonstrate DNA helicase/translocase activity. These experiments need to be carried out. These experiments can also be used to clarify the polarity issues raised by the reviewers.*

Both the editor and some of the reviewers appear to expect that RadA branch-migrates by a similar mechanism to RecG and RuvAB. An alternative, not considered by them, is that RadA uses a RecA-like strand transfer mechanism to branch migrate DNA. Because of the sequence similarity of RadA to RecA, including highly conserved residues in regions that align to primary and secondary DNA binding sites in RecA, we think that a RecA-like mechanism should be seriously considered. RecA does not act as a translocase or helicase and our failure to find RadA helicase activity may be consistent with a RecA-like mechanism. This idea has been added to the discussion, where we discuss both DNA pump and strand-transfer branch migration mechanisms. We note, in addition, that RecA does not have special affinity to branched structures, which may be true for RadA as well. RadA shows no structure-specific stimulation of its ATPase; this contrasts to RuvAB and RecG, which do show structure-specific stimulation of ATPase activity. We therefore disagree that demonstration of translocase or helicase activity should be a condition for publication – RadA may simply use another mechanism.

Further experiments clearly should be done to clarify the mechanism but we feel that this is beyond the scope of this work.

*All three reviewers have extensive and specific comments that should be addressed. While further experimentation will take some time to complete, I am sure that it will be worthwhile and lead to significant improvements to the manuscript. Reviewer #1: This manuscript describes the biochemical properties of the E. coli RadA protein. While potentially interesting, the manuscript disappoints due to the poor (or incomplete) quality of some of the datasets. The interpretation of several experiments is also questionable. Some examples: DNA binding and ATPase assays: The work suggests that RadA binds to poly(dT) but not poly(dA), (dC) or (dG). Yet these (non-bound) polynucleotides stimulate the ATPase activity of RadA indicating that they do in fact interact with RadA. Presumably the interaction with poly(dT), observed by PAGE is a reflection of the stability of binding rather than binding per se* – *especially since binding is only seen in the presence of ADP. The turnover rates on the other substrates (and with ATP) may be significantly greater than that observed with poly(dT). Have attempts been made to stabilize complexes by glutaraldehyde fixation?*

Yes, with no improvement.

*Also, from the gel data provided it is not clear whether the observed binding band represents a defined RadA-DNA complex or an aggregate that fails to enter the gel.*

On other gels, it does enter (such as Figure 1—figure supplement 4). Our His-tagged protein, not as active as this version, produced a ladder of shifted species.

*Binding assays would benefit from the use of quantitative assays such as fluorescence anisotropy or surface plasmon resonance, rather than gel assays. Given that RadA appears to drive junction branch migration, the assays should be carried out with single-stranded (both defined and random sequence), duplex, 3-arm and 4-arm (Holliday junction) DNA substrates. Only then will the substrate specificity become apparent. Overall, the sections on DNA binding and ATPase activity are naïve and let the manuscript down.*

Unfortunately, we have no experience with fluorescence anisotropy and have no equipment for surface plasmon resonance. Moreover, we think that the outcome would almost certainly be uninteresting. We have investigated branched structure stimulation of ATPase assays, now presented in Figure 2—figure supplement 1. As stated above, RadA does not appear to have special affinity for any branched structure – it seems to like single-stranded DNA. This would be consistent with a RecA-like mechanism.

*Branch migration assays: Some of the data showing that RadA promotes branch migration is of low quality. For example, Figure 3 looks completely different from the other figure panels, and I find it difficult to conclude anything from the data of Figure 5.*

The branch migration figure has been redone and quantitated in Figure 6. The directionality data are quantitated and expressed with error bars. This experiment has been performed with three different intermediate preps and two different protein preps and the result is consistent and clear. The other figure (what reviewer 1 calls 3C but I think is 4C), an order of addition experiment has been redone as a time course and is now in Figure 4—figure supplement 1. Reviewer 1 should find both of these to be more convincing.

*The 4-strand reactions shown in Figure 6 are also problematic, since there appears to be a substantial amount of contaminating single-stranded DNA (SCS) present. Or is this experiment a comparison of the 3- and 4-strand reaction (the legend of the time course does not say what the difference is between the first 4 and the second 4 lanes). As a reader I can only guess. Also, interpretation of these experiments, and the 3-strand reaction described earlier would greatly benefit from the use of 32P-labels. Incidentally, what is a 1.2 gap (as stated in legend).*

A label disappeared upon conversion of one of the files – this figure (now Figure 7) indeed shows both 3-strand and 4-strand reactions performed in parallel. We apologize for this error, as we realize it must have been frustrating. The interpretation of this figure should now be very clear. The gap is 1346 nucleotides, corrected. The invading end of the linear would anneal to about 2/3 of the gap. Note the reproducibility of 3-strand reaction with RadA RecA vs. RecA.

*Other enzymes that promote branch migration, such as RuvAB and RecG, are DNA translocases/helicases. To make the argument that RadA is a true branch migration protein this needs to be demonstrated for RadA.*

As mentioned above, we think that RadA could branch migrate DNA via a strand transfer mechanism like RecA and not be a translocase/helicase protein. We hope to have clarified this point in the manuscript for the readers as well. By the way, we note that specific features of RecA filament are believed to be used in the homology-search process and are not used in the strand exchange process. Therefore, it is conceivable that RadA could be a RecA-like molecule that is poor at homology searching and initial pairing, but very good at strand transfer and branch migration.

*Polarity: The directionality of RecG is 3'-5' relative to single stranded DNA, whereas RuvAB is 5'-3' (the same as RecA). These helicase polarities translate as directionality in the branch migration reactions they promote (at least with 3-stranded reactions). The reader is led to assume that RadA is a translocase similar to RecG or RuvAB* – *if so then the authors should show it and determine the polarity (rather than propose* – *without any evidence* –

*that there is a polarity in the way that RadA is recruited to RecA filaments).*

Again, RadA is not necessarily a helicase/translocase. We do not know whether this polarity reflects RadA:RadA interactions or some preference for 5’ flaps vs. 3’ flaps. We hope to clarify this in the future by EM studies of RadA structures on DNA but think this is beyond the scope of this report.

*Summary: While the manuscript has some aspects that are potentially interesting, much of the data provided is not of the standard expected for publication in eLife. Also, essential experiments to convince the reader that this is a true branch migration protein are missing* –

*the authors need to demonstrate that RadA shows specific binding to DNA substrates containing 3- or 4-way junctions, and to show that the protein is a DNA translocase (helicase). Without such data the work is simply too preliminary for publication in eLife.*

We think the new data and their presentation are now up to the *eLife* standard. We think we have convincingly demonstrated a branch migration activity for RadA, a known recombination factor with heretofore-unknown function. As such, this represents an important contribution to the field of recombination. The fact that this is a strand-exchange protein paralog is particularly intriguing. It represents a new paradigm for paralog function and a potentially interesting mechanism, with lots of biological implications, for facilitating branch migration. This reports the first biochemical analysis of a protein that is an absolutely conserved protein in bacteria and plants. There were many publications on RecG and RuvAB before their mechanism was fully elucidated and while we would like to know RadA’s mechanism, more work needs to be done to solve this problem.

*Reviewer #2: The bacterial RecA protein plays an important role in the reconstitution of collapsed replication forks, particularly those in which one arm of the fork has been detached as a double strand break. The reactions promoted by RecA can carry out the key reactions needed to reconstruct the fork. However, there has always been a problem. RecA-promoted DNA strand exchange has been too slow to account for the rates of fork reactivation observed in vivo. In this important work, Dr. Lovett and colleagues provide the first in vitro characterization of the E. coli RadA protein, a protein that likely solves the strand exchange rate problem. The authors convincingly demonstrate that the addition of RadA to RecA reactions provides an impressive boost to the rates of many RecA reactions, complementing RecA in these endeavors. Although the work opens many new questions about how the rate enhancements are brought about, the overall effort is an extremely important step and provides a window into the (until recently) obscure bacterial RadA protein that should stimulate much additional work. A series of thoughtfully designed point mutants also reveal that the key RadA activities are dependent on the ATP hydrolysis, Zn finger, and unique KNRFG motifs of RadA. While these data provide a great addition to our knowledge of homologous recombination, the following issues need to be addressed: The only RadA concentrations used in this study when RecA is present appear to be those needed to effect a 1:17 ratio with RecA (subsection “Strand exchange reactions”, first paragraph; Figure 4 legend). The reasons for this ratio, or results from other concentrations of RadA, are never presented. Some information should be provided as to why this ratio was chosen and what happens if other RadA concentrations are used. Are the effects of RadA saturated at these concentrations? Can lower levels of RadA suffice? Such information might provide a clue as to how processive or distributive the RadA function is.*

We now include a titration of RadA in RecA reactions, Figure 4—figure supplement 3 and the results discussed in the text. As you can now see, we chose the ratio as one with almost maximal stimulation of RecA. (RecA is included at saturating concentration). Lower levels can indeed suffice, but produce products at a slower rate, suggesting a distributive effect. We see no evidence of a “threshold” effect.

*The authors appropriately highlight the fact that RadA migrates the junction in the same direction as RecA (Abstract and end of Introduction). However, the presence of SSB, which will be ubiquitous in the cell, reverses this directionality in experiments where RecA is removed prior to RadA addition (Figure 5). The implications and context of this change in directionality are not discussed.*

We show in the new figure that the directionality is lost rather than reversed. We do not understand the basis for this – which might include how well SSB packs at 3’ vs. 5’ flaps and I can’t find any information in the literature about this. I do believe this, and the SSB inhibition of RadA, has strong biological implications and may prevent RadA from reversing recombination reactions, products in which the invading strand abuts SSB-bound ssDNA. I have briefly aluded to this in the discussion of the role of branch migration in recombination.

*What is the difference between the DNA substrates in Figure 4 and those in Figure 4—figure supplement 1? The absence of 0 timepoints in Figure 4—figure supplement 1 makes it hard to figure out what is what.*

There is no difference; this is an order-of-addition experiment. This figure has been replaced with a time course, new Figure 4—figure supplement 1, which is more clearly labeled and includes a time course rather than a single point in the reaction.

*In Figure 4, perhaps the lower yield of nicked DNA products when RadA is used with RecA K72R is worth commenting on.*

Because these are different reaction conditions, it is difficult to interpret. This mutant also makes intermediates more slowly and RadA appears to require intermediates to stimulate exchange.

*Reviewer #3: Genetic evidence supports role for RadA in E. coli in the late steps in recombination, potentially involving branch migration of DNA recombination intermediates. In this manuscript, the authors purified wild type RadA and several mutant proteins, and showed that RadA increases the rate of RecA-mediated recombination in vitro by stimulation of branch migration. Authors also showed that RadA preferentially binds single-strand DNA in the presence of ADP and that it exhibits ATPase activity stimulated by DNA. The results are interesting and novel, and given that E. coli recombination has been very extensively studied, the findings are particularly unexpected. The general finding that RadA can convert 3-stranded joint molecule intermediates into Holliday junctions is important.*

We think so too and think this makes it suitable for *eLife* publication.

That said, the manuscript is not without its flaws. The presentation of some data in some figures is not well done, and has the feeling more of a lab report, rather than manuscript written for a broad readership. Finally, putting editorial and presentation issues aside, the results presented do support the authors' conclusions. However, the choice of some of the experiments that were done is at best quirky, and in some cases, stopping just short of what could have/should have been done.

We think the presentation is much improved, having incorporated Reviewer 3’s suggestions, please see below.

*Knowing that RadA promotes migration of 3-way and 4-way junctions, the DNA binding and ATP hydrolysis experiments in this manuscript stop short of supporting the proposed the branch migration function of RadA during recombination: they fall into the category of basic characterization. Perhaps though not essential, the authors should have tested 4-way junctions or branched DNA substrates in their DNA binding and ATPase activity assays: they only tested ssDNA and dsDNA and consequently those data do not directly address the function of RadA during branch migration.*

We don’t see strong binding (and we have not developed a method of visualizing weak/transient binding) but do see stimulation of ATPase activity, new Figure 2—figure supplement 1, but not any more than ssDNA. I don’t think that showing branch migration of oligonucleotide substrates would add much to the paper – Figure 7 now clearly demonstrates branch migration of polynucleotide substrates. So RadA IS a branch-migration protein. Again, we don’t see any differential response to branched DNA – this may just be the way it is (as for RecA) or we may be missing something (conditions, cofactor or protein) that enhances binding to such structures. We can show a gel shift for a ssDNA oligonucleotide, natural composition, at higher RadA concentrations if that would help, but again that is not going to add much to how we understand RadA. We are not going to be able to demonstrate tighter binding to junctions – it just isn’t there – and this property is not necessarily required for catalysis of branch migration.

*Furthermore, in the competition experiments, they showed that RadA binds to ssDNA especially poly dT, but did not test any branched structures. Enigmatically, they showed that phiX174 ssDNA stimulated the ATPase of RadA, although RadA apparently does not bind to this DNA in mobility shift experiment ("data not shown"). These results do not help to understand the branch migration function of RadA.*

RadA doesn’t bind oligonucleotide structures well but they do stimulate its ATPase (now shown in Figure 2—figure supplement 1). Since we can clearly demonstrate DNA-stimulated ATPase, we infer DNA binding, of a transient or unstable nature under the same conditions.

*The choice to present the data in Figure 2 as essentially raw data without any labels makes it nearly unintelligible. The authors should relegate this figure (corrected as indicated below) to the supplementary info, and instead provide panel A as a table or bar graph; B, as an x-y graph, and C as a table or bar graph. The same comment applies to Figure 3: The authors should relegate this figure to the supplementary info, and instead provide panels A and B as a table or bar graph.*

We thank Reviewer 3 for these suggestions, and now present the data both in Table 1 and bar graphs in Figure 2. Labels should also be more clear.

*More specific comments: In the Abstract, the authors say "RadA (Sms)" but there is no explanation for what "Sms" is even in the main text and its relationship to RadA.*

It stands for “Sensitivity to MMS”, now explained and cited. We tend to use the RadA/Sms designation to distinguish it from archaeal RadA, an unfortunate nomenclature that we are stuck with.

*In the Abstract it is stated: "Unlike other branch migration factors RecG and RuvAB, RadA stimulates branch migration in the direction of RecA-mediated strand exchange", and in the Discussion "In contrast, when purified RuvAB or RecG are added to RecA-strand transfer reactions, they decrease the recovery of full strand-exchange products, by accelerating the reverse reaction back to substrate forms (Whitby et al., 1993)." However, RuvAB can branch migrate in the same direction as RecA. In the paper cited by authors, (Witby et al., 1993), RuvAB was shown to stimulate RecA-mediated 4-strand exchange to produce final nicked product. Other papers also showed that RuvAB-mediated branch migration direction can be opposite to RecG or bidirectional (Matthew and Lloyd (1995) EMBO-J 14 3302-3310; Iwasaki et al. (1992) Genes Dev. 6 2214-2220; Tsaneva et al. (1992) Cell 69 1171-1180).*

Indeed, but neither protein can branch migrate in a RecA-coupled reaction towards products. These intrinsic polarities are now discussed and cited, to make it clear.

*In the second paragraph of the Introduction: "ATP binding, but not hydrolysis, is required for RecA filament formation." This should be changed to "active filament" formation. RecA can make filaments in the absence of nucleotide, though it is compressed and inactive.*

True, fixed.

*Introduction, sixth paragraph and onward: It will be very helpful for readers if a schematic of RadA, showing the three conserved domains, were provided.*

Done. This came out nicely.

*Figure 1—figure supplement 1: There is no indication of the molecular weights of marker proteins. The contents of each lane are not described. The gel of lane Q does not have molecular weight marker. Also, please indicate with an arrow which bands in the gel are the relevant bands.*

Done.

*RadA mutant purification: The authors purified RadA wild type and C28Y (Zn finger), K108R (Walker A box), K258A (KNRFG RadA motif), and S372A (putative Lon protease active site) mutant proteins. However, they only showed wild type protein in Figure 1—figure supplement 1. They need to show all of their purified proteins that were used in this manuscript.*

Done.

*Figure 1—figure supplement 2. The gel in this figure needs to be quantified. A graph with the fitted Kd needs to be shown.*

Done. Also derived Hill Coefficient.

*Figure 1—figure supplement 3* – *It would be most helpful to the reader if the authors showed the DNA substrates in diagram and which DNA is labeled with 32P.*

This has been replaced.

*In the second paragraph of the subsection “DNA binding”: The sources and average lengths of poly(dT), poly(dA), poly(dG) and poly(dC) are not provided.*

Done. These are defined oligos.

*In the second paragraph of the subsection “DNA binding”: The authors concluded that the binding affinity is 90 nM. However, the molecule concentration of p(dT)_30_ was 11 nM, so the nucleotide concentration is 330 nM. Therefore, the result in Figure 1—figure supplement 2 may show that the stoichiometry between RadA and ssDNA is 1:2 (~180:330) but not that the binding affinity is 90 nM. To determine the binding affinity, one should use the less concentration of substrate than the binding affinity.*

I’m afraid we don’t understand this last statement.

*In the second paragraph of the subsection “DNA binding” and entire manuscript: "substrate E2" and "substrate E-2" are mixed in the text.*

Corrected.

*In the subsection “DNA-stimulated ATPase”: "these DNA molecules are not bound by RadA detectably in gel shift experiments (data not shown)". Were these experiments done in the presence of ATP or ADP?*

Both, plus AMP, dATP, dADP, AMP-PNP, ATPgammaS, etc.

*Figure 2: The symbols of plots are not clear in the figure. The authors must use larger symbols and color is recommended.*

Symbols are larger, color has been added, and lines are labeled.

*Figure 4: Strand exchange reactions. The experimental protocol is confusing in the manuscript. In the subsection “Strange exchange reactions”, they stated "Presynaptic filaments are formed by incubation of RecA with ssDNA in the presence of ATP, SSB is then added and the reaction is initiated by the addition of the linear dsDNA". But then the Materials and methods "6.7 μM RecA and RadA as indicated (most typically 400 nM) was incubated with 20.1 μM viral φX174 DNA f 436 or 8 minutes at 37o. Then, 20.1 μM linear double-strand φX174 DNA was added and incubation was continued for 5 minutes. Reactions were initiated by addition of ATP to 3 mM and SSB to 2.1 μM". However, in the figure legend, "the standard procedure for the reaction is: 1) Incubation at 37 for 8 minutes of buffer, ATP regenerating system, φX174 single-strand DNA, and RecA (and RadA when included). 2) Addition of double-strand linear φX174 and continued incubation for 5 minutes. 3) Addition of ATP and SSB and incubation for the times indicated. Addition of SSB and ATP is critical in this experiment because RecA bind to ATP and RecA also competes with SSB for DNA binding during experiment." It is essential that the authors clearly and unambiguously define their experimental conditions.*

This has been corrected. Figure 4 and Figure 4—figure supplement 1 shows that RadA stimulates RecA reactions with dsDNA or SSB/ATP added either in step 2 or 3.

*Figure 4: "Lane 2-Standard order with no RadA" Why hasn't the joint molecule accumulated after 15 min? In Figure 4, the linear dsDNA band disappeared at 15 min and was completed converted to intermediate and a bit of nicked product, with RecA but without RadA.*

This figure has been replaced and is easier to understand.

*Figure 4: "Lane 3-Stand order with RadA" -> "Lane 3-Standard order with RadA".*

Done.

*In the subsection “Strand exchange reactions”: "RecA-mediated strand exchange at 5 minutes" According to the figure legend, the incubation time was 15 min. Is 5 or 15 min correct?*

This has been replaced.

*In the subsection “Strand exchange reactions”: "we performed standard RecA strand-exchange reactions for 15 min and purified the DNA from such reactions after protease treatment". According to figure legend, the incubation time with RecA was 12 min. Is 12 or 15 min correct?*

12. This has been corrected.

*Figure 5: Has this experiment been reproduced? There are no error bars. Also, add labels to the lines, and use different colors for each data set as appropriate.*

Replaced with new Figure 6. This experiment has been done with 3 different substrate preps and 2 different protein preps. Directionality data are now averaged and shown.

*Figure 6: There are two sets of experiments for both RecA only and RecA-RadA reactions. What was the difference in reaction conditions for each half of each reaction?*

These are 3-strand vs. 4-strand, now clearly labeled.

*In the subsection “Strand exchange reactions, sixth paragraph: "D108R" -> "K108R".*

Done.

*In the subsection “Strand exchange reactions”, last paragraph and in the Discussion, fourth paragraph: The experimental data supporting the 3-strand reaction in the presence of ADP is apparently not shown in the manuscript. This paragraph must be deleted, or the corresponding data needs to be presented.*

Data are now presented as part of Figure 4.

*Discussion, fifth paragraph: "true recombinases" -> "DNA strand exchange proteins".*

Replaced.

Figure 6 legend: "with a 1.2 single-strand gap". What is 1.2? knt?

Replaced with 1346 nucleotide gap.